# Personalized Federated Learning with Moreau Envelopes

**Canh T. Dinh**[1], **Nguyen H. Tran**[1], **Tuan Dung Nguyen**[1,2]

[1]The University of Sydney, Australia
`tdin6081@uni.sydney.edu.au`, `nguyen.tran@sydney.edu.au`
[2]The University of Melbourne, Australia
`tuandungn@unimelb.edu.au`

## Abstract

Federated learning (FL) is a decentralized and privacy-preserving machine learning technique in which a group of clients collaborate with a server to learn a global model without sharing clients' data. One challenge associated with FL is statistical diversity among clients, which restricts the global model from delivering good performance on each client's task. To address this, we propose an algorithm for personalized FL (pFedMe) using Moreau envelopes as clients' regularized loss functions, which help decouple personalized model optimization from the global model learning in a bi-level problem stylized for personalized FL. Theoretically, we show that pFedMe's convergence rate is state-of-the-art: achieving quadratic speedup for strongly convex and sublinear speedup of order 2/3 for smooth non-convex objectives. Experimentally, we verify that pFedMe excels at empirical performance compared with the vanilla FedAvg and Per-FedAvg, a meta-learning based personalized FL algorithm.

## 1 Introduction

The abundance of data generated in a massive number of hand-held devices these days has stimulated the development of Federated learning (FL) [1]. The setting of FL is a network of clients connected to a server, and its goal is to build a global model from clients' data in a privacy-preserving and communication-efficient way. The current techniques that attempt to fulfill this goal mostly follow three steps: (i) at each communication iteration, the server sends the current global model to clients; (ii) the clients update their local models using their local data; (iii) the server collects the latest local models from a subset of sampled clients in order to update a new global model, repeated until convergence [1–4].

Despite its advantages of data privacy and communication reduction, FL faces a main challenge that affects its performance and convergence rate: statistical diversity, which means that data distributions among clients are distinct (i.e., non-i.i.d.). Thus, the global model, which is trained using these non-i.i.d. data, is hardly well-generalized on each client's data. This particular behaviour has been reported in [5, 6], which showed that when the statistical diversity increases, generalization errors of the global model on clients' local data also increase significantly. On the other hand, individual learning without FL (i.e., no client collaboration) will also have large generalization error due to insufficient data. These raise the question: *How can we leverage the global model in FL to find a "personalized model" that is stylized for each client's data?*

Motivated by critical roles of personalized models in several business applications of healthcare, finance, and AI services [5], we address this question by proposing a new FL scheme for personalization, which minimizes the Moreau envelopes [7] of clients' loss functions. With this scheme,

clients not only contribute to building the "reference" global model as in the standard FL, but also leverage the reference model to optimize their personalized models w.r.t. local data. Geometrically, the global model in this scheme can be considered as a "central point" where all clients agree to meet, and personalized models are the points in different directions that clients follow according to their heterogeneous data distributions.

**Our key contributions** in this work are summarized as follows. First, we formulate a new bi-level optimization problem designed for personalized FL (pFedMe) by using the Moreau envelope as a regularized loss function. The bi-level structure of pFedMe has a key advantage: decoupling the process of optimizing personalized models from learning the global model. Thus, pFedMe updates the global model similarly to the standard FL algorithm such as FedAvg [1], yet parallelly optimizes the personalized models with low complexity.

Second, we exploit the convexity-preserving and smoothness-enabled properties of the Moreau envelopes to facilitate the convergence analysis of pFedMe, which characterizes both client-sampling and client-drift errors: two notorious issues in FL [3]. With carefully tuned hyperparameters, pFedMe can obtain the state-of-the-art quadratic speedup (resp. sublinear speedup of order $2/3$), compared with the existing works with linear speedup (resp. sublinear speedup of order $1/2$), for strongly convex (resp. smooth nonconvex) objective.

Finally, we empirically evaluate the performance of pFedMe using both real and synthetic datasets that capture the statistical diversity of clients' data. We show that pFedMe outperforms the vanilla FedAvg and a meta-learning based personalized FL algorithm Per-FedAvg [8] in terms of convergence rate and local accuracy.

## 2   Related Work

**FL and challenges.** One of the first FL algorithms is FedAvg [1], which uses local SGD updates and builds a global model from a subset of clients with non-i.i.d. data. Subsequently, one-shot FL [9] allows the global model to learn in one single round of communication. To address the limitations on communications in a FL network, [10, 11] introduced quantization methods, while [12–14] proposed performing multiple local optimization rounds before sending the local models to the server. In addition, the problem of statistical diversity has been addressed in [15–20]. Preserving privacy in FL has been studied in [21–25].

**Personalized FL: mixing models, contextualization, meta-learning, and multi-task learning.** Multiple approaches have been proposed to achieve personalization in FL. One such approach is *mixing* the global and local models. [26] combined the optimization of the local and global models in its L2GD algorithm. [27] introduced three personalization approaches to: user clustering, data interpolation, and model interpolation. While the first two approaches need meta-features from all clients that make them not feasible in FL due to privacy concern, the last approach was used in [6] to create an adaptive personalized federated learning (APFL) algorithm, which attempted to mix a user's local model with the global model. One personalization method used in neural networks is FedPer [28], in which a network is divided into base and personalized layers, and while the base layers are trained by the server, both types of layers will be trained by users to create a personalized model. Regarding using a model in different *contexts*, in the next-character prediction task in [29], the requirement to predict differently among devices raises a need to inspect more features about the context of client devices during training, which was studied in [30]. [31] achieves personalization on each user in a fully decentralized network using asynchronous gossip algorithms with assumptions on network topology and similarity between users. The concept of personalization can also be linked to *meta-learning*. Per-FedAvg [8], influenced by Model-Agnostic Meta-Learning (MAML) [32], built an initial meta-model that can be updated effectively after one more gradient descent step. During meta-optimization, however, MAML theoretically requires computing the Hessian term, which is computationally prohibitive; therefore, several works including [32–34] attempted to approximate the Hessian matrix. [35] based its framework, ARUBA, on online convex optimization and meta-learning, which can be integrated into FL to improve personalization. [36] discovered that FedAvg can be interpreted as meta-learning and proposed combining FedAvg with Reptile [33] for FL personalization. The application of federated meta-learning in recommender systems was studied in [37]. Finally, *multi-task learning* can be used for personalization: [20] introduced a federated multi-task framework

called MOCHA, addressing both systems and statistical heterogeneity. For more details about FL, its challenges, and personalization approaches, we refer the readers to comprehensive surveys in [38, 39].

## 3  Personalized Federated Learning with Moreau Envelopes (pFedMe)

### 3.1  pFedMe: Problem Formulation

In conventional FL, there are $N$ clients communicating with a server to solve the following problem:

$$\min_{w \in \mathbb{R}^d}\Big\{ f(w) := \frac{1}{N}\sum_{i=1}^{N} f_i(w)\Big\} \tag{1}$$

to find a *global model* $w$. The function $f_i : \mathbb{R}^d \to \mathbb{R}$, $i = 1, \dots, N$, denotes the expected loss over the data distribution of the client $i$:

$$f_i(w) = \mathbb{E}_{\xi_i}\big[\tilde{f}_i(w; \xi_i)\big],$$

where $\xi_i$ is a random data sample drawn according to the distribution of client $i$ and $\tilde{f}_i(w; \xi_i)$ is a loss function corresponding to this sample and $w$. In FL, since clients' data possibly come from different environments, contexts, and applications, clients can have non-i.i.d. data distributions, i.e., the distributions of $\xi_i$ and $\xi_j$, $i \neq j$, are distinct.

Instead of solving the traditional FL problem (1), we take a different approach by using a regularized loss function with $l_2$-norm for each client as follows

$$f_i(\theta_i) + \frac{\lambda}{2}\|\theta_i - w\|^2, \tag{2}$$

where $\theta_i$ denotes the *personalized model* of client $i$ and $\lambda$ is a regularization parameter that controls the strength of $w$ to the personalized model. While large $\lambda$ can benefit clients with unreliable data from the abundant data aggregation, small $\lambda$ helps clients with sufficient useful data prioritize personalization. Note that $\lambda \in (0, \infty)$ to avoid extreme cases of $\lambda = 0$, i.e., no FL, or $\lambda = \infty$, i.e., no personalized FL. Overall, the idea is allowing clients to pursue their own models with different directions, but not to stay far away from the "reference point" $w$, to which every client contributes. Based on this, the personalized FL can be formulated as a bi-level problem:

$$\texttt{pFedMe}: \quad \min_{w \in \mathbb{R}^d}\Big\{ F(w) := \frac{1}{N}\sum_{i=1}^{N} F_i(w)\Big\}, \text{ where } F_i(w) = \min_{\theta_i \in \mathbb{R}^d}\Big\{ f_i(\theta_i) + \frac{\lambda}{2}\|\theta_i - w\|^2\Big\}.$$

In pFedMe, while $w$ is found by exploiting the data aggregation from multiple clients at the outer level, $\theta_i$ is optimized with respect to (w.r.t) client $i$'s data distribution and is maintained a bounded distance from $w$ at the inner level. The definition of $F_i(w)$ is the well-known Moreau envelope, which facilitates several learning algorithm designs [40, 41]. The optimal personalized model, which is the unique solution to the inner problem of pFedMe and also known as the proximal operator in the literature, is defined as follows:

$$\hat{\theta}_i(w) := \operatorname{prox}_{f_i/\lambda}(w) = \operatorname*{arg\,min}_{\theta_i \in \mathbb{R}^d}\Big\{ f_i(\theta_i) + \frac{\lambda}{2}\|\theta_i - w\|^2\Big\}. \tag{3}$$

For comparison, we consider Per-FedAvg [8], which arguably has the closest formulation to pFedMe:

$$\min_{w \in \mathbb{R}^d}\Big\{ F(w) := \frac{1}{N}\sum_{i=1}^{N} f_i\big(\theta_i(w)\big)\Big\}, \text{ where } \theta_i(w) = w - \alpha\nabla f_i(w). \tag{4}$$

Based on the MAML framework [32], Per-FedAvg aims to find a global model $w$ which client $i$ can use as an *initialization* to perform one more step of gradient update (with step size $\alpha$) w.r.t its own loss function to obtain its personalized model $\theta_i(w)$.

Compared to Per-FedAvg, our problem has a similar meaning of $w$ as a "meta-model", but instead of using $w$ as the initialization, we, in parallel, pursue both the personalized and global models by solving a bi-level problem, which has several benefits. First, while Per-FedAvg is optimized for one-step gradient update for its personalized model, pFedMe is agnostic to the inner optimizer, which means (3) can be solved using any iterative approach with multi-step updates. Second, by re-writing the personalized model update of Per-FedAvg as

$$\theta_i(w) = w - \alpha\nabla f_i(w) = \operatorname*{arg\,min}_{\theta_i \in \mathbb{R}^d}\Big\{ \langle \nabla f_i(w), \theta_i - w\rangle + \frac{1}{2\alpha}\|\theta_i - w\|^2\Big\}, \tag{5}$$

where we use $\langle x, y \rangle$ for the inner product of two vectors $x$ and $y$, we can see that apart from the similar regularization term, Per-FedAvg only optimizes the first-order approximation of $f_i$, whereas pFedMe directly minimizes $f_i$ in (3). Third, Per-FedAvg (or generally several MAML-based methods) requires computing or estimating Hessian matrix, whereas pFedMe only needs gradient calculation using first-order approach, as will be shown in the next section.

**Assumption 1** (Strong convexity and smoothness). *$f_i$ is either (a) $\mu$-strongly convex or (b) nonconvex and $L$-smooth (i.e., $L$-Lipschitz gradient), respectively, as follows when $\forall w, w'$:*

$$(a)\ f_i(w) \geq f_i(w') + \langle \nabla f_i(w'), w - w' \rangle + \frac{\mu}{2} \|w - w'\|^2,$$

$$(b)\ \|\nabla f_i(w) - \nabla f_i(w')\| \leq L \|w - w'\|.$$

**Assumption 2** (Bounded variance). *The variance of stochastic gradients in each client is bounded*

$$\mathbb{E}_{\xi_i}\left[\|\nabla \tilde{f}_i(w; \xi_i) - \nabla f_i(w)\|^2\right] \leq \gamma_f^2.$$

**Assumption 3** (Bounded diversity). *The variance of local gradients to global gradient is bounded*

$$\frac{1}{N} \sum\nolimits_{i=1}^{N} \|\nabla f_i(w) - \nabla f(w)\|^2 \leq \sigma_f^2.$$

While Assumption 1 is standard for convergence analysis, Assumptions 2 and 3 are widely used in FL context in which $\gamma_f^2$ and $\sigma_f^2$ quantify the sampling noise and the diversity of client's data distribution, respectively [3, 8, 42, 43]. Note that we avoid using the uniformly bounded gradient assumption, i.e., $\|\nabla f_i(w)\| \leq G, \forall i$, which was used in several related works [6, 8]. It was shown that this assumption is not satisfied in the unconstrained strongly convex minimization [44, 45].

Finally, we review several useful properties of the Moreau envelope such as smoothing and preserving convexity as follows (see the review and proof for the convex case in [40, 46, 47] and for nonconvex smooth case in [48], respectively):

**Proposition 1.** *If $f_i$ is convex or nonconvex with $L$-Lipschitz $\nabla f_i$, then $\nabla F_i$ is $L_F$-smooth with $L_F = \lambda$ (with the condition that $\lambda > 2L$ for nonconvex $L$-smooth $f_i$), and*

$$\nabla F_i(w) = \lambda(w - \hat{\theta}_i(w)). \tag{6}$$

*Furthermore, if $f_i$ is $\mu$-strongly convex, then $F_i$ is $\mu_F$-strongly convex with $\mu_F = \frac{\lambda\mu}{\lambda+\mu}$.*

### 3.2 pFedMe: Algorithm

In this section, we propose an algorithm, presented in Alg. 1, to solve pFedMe. Similar to conventional FL algorithms such as FedAvg [1], at each communication round $t$, the server broadcasts the latest global model $w_t$ to all clients. Then, after all clients perform $R$ local updates, the server will receive the latest local models from a uniformly sampled subset $\mathcal{S}^t$ of clients to perform the model averaging. Note that we use an additional parameter $\beta$ for global model update, which includes FedAvg's model averaging when $\beta = 1$. Though a similar parameter at the server side was also used in [3, 49], it will be shown that pFedMe can obtain better speedup convergence rates.

Specifically, our algorithm, which aims to solve the bi-level problem pFedMe, has two key differences compared with FedAvg, which aims to solve (1). First, at the inner level, each client $i$ solves (3) to obtain its personalized model $\hat{\theta}_i(w_{i,r}^t)$ where $w_{i,r}^t$ denotes the *local model* of the client $i$ at the global round $t$ and local round $r$. Similar to FedAvg, the purpose of local models is to contribute to building a global model with reduced communication rounds between the clients and server. Second, at the outer level, the local update of client $i$ using gradient descent is with respect to $F_i$ (instead of $f_i$) as the following

$$w_{i,r+1}^t = w_{i,r}^t - \eta \nabla F_i\left(w_{i,r}^t\right),$$

where $\eta$ is the learning rate and $\nabla F_i\left(w_{i,r}^t\right)$ is calculated according to (6) using the current personalized model $\hat{\theta}_i(w_{i,r}^t)$.

For the practical algorithm, we use a $\delta$-approximation of $\hat{\theta}_i(w_{i,r}^t)$, denoted by $\tilde{\theta}_i(w_{i,r}^t)$ satisfying $\mathbb{E}\left[\|\tilde{\theta}_i(w_{i,r}^t) - \hat{\theta}_i(w_{i,r}^t)\|\right] \leq \delta$, and correspondingly use $\lambda(w_{i,r}^t - \tilde{\theta}_i(w_{i,r}^t))$ to approximate $\nabla F_i(w_{i,r}^t)$

---

**Algorithm 1** pFedMe: Personalized Federated Learning using Moreau Envelope Algorithm

---
1: **input:** $T$, $R$, $S$, $\lambda$, $\eta$, $\beta$, $w^0$
2: **for** $t = 0$ to $T - 1$ **do**                                             ▷ Global communication rounds
3:     Server sends $w_t$ to all clients
4:     **for** all $i = 1$ to $N$ **do**
5:         $w_{i,0}^t = w_t$
6:         **for** $r = 0$ to $R - 1$ **do**                                  ▷ Local update rounds
7:             Sample a fresh mini-batch $\mathcal{D}_i$ with size $|\mathcal{D}|$ and minimize $\tilde{h}_i(\theta_i; w_{i,r}^t, \mathcal{D}_i)$, defined in (7), up to an accuracy level according to (8) to find a $\delta$-approximate $\tilde{\theta}_i(w_{i,r}^t)$
8:             $w_{i,r+1}^t = w_{i,r}^t - \eta\lambda(w_{i,r}^t - \tilde{\theta}_i(w_{i,r}^t))$
9:     Server uniformly samples a subset of clients $\mathcal{S}^t$ with size $S$, and each of the sampled client sends the local model $w_{i,R}^t, \forall i \in \mathcal{S}^t$, to the server
10:    Server updates the global model: $w_{t+1} = (1 - \beta)w_t + \beta \sum_{i \in \mathcal{S}^t} \frac{w_{i,R}^t}{S}$

---

(c.f. line 8). The reason for using the $\delta$-approximate $\tilde{\theta}_i(w_{i,r}^t)$ is two-fold. First, obtaining $\hat{\theta}_i(w_{i,r}^t)$ according to (3) usually needs the gradient $\nabla f_i(\theta_i)$, which, however, requires the distribution of $\xi_i$. In practice, we use the following unbiased estimate of $\nabla f_i(\theta_i)$ by sampling a mini-batch of data $\mathcal{D}_i$

$$\nabla \tilde{f}_i(\theta_i, \mathcal{D}_i) := \frac{1}{|\mathcal{D}_i|} \sum_{\xi_i \in \mathcal{D}_i} \nabla \tilde{f}_i(\theta_i; \xi_i)$$

such that $\mathbb{E}[\nabla \tilde{f}_i(\theta_i, \mathcal{D}_i)] = \nabla f_i(\theta_i)$. Second, in general, it is not straightforward to obtain $\hat{\theta}_i(w_{i,r}^t)$ in closed-form. Instead we usually use iterative first-order approach to obtain an approximate $\tilde{\theta}_i(w_{i,r}^t)$ with high accuracy. Defining

$$\tilde{h}_i(\theta_i; w_{i,r}^t, \mathcal{D}_i) := \tilde{f}_i(\theta_i; \mathcal{D}_i) + \frac{\lambda}{2}\|\theta_i - w_{i,r}^t\|^2, \tag{7}$$

suppose we choose $\lambda$ such that $\tilde{h}_i(\theta_i; w_{i,r}^t, \mathcal{D}_i)$ is strongly convex with a condition number $\kappa$ (which quantifies how hard to optimize (7)), then we can apply gradient descent (resp. Nesterov's accelerated gradient descent) to obtain $\tilde{\theta}_i(w_{i,r}^t)$ such that

$$\|\nabla \tilde{h}_i(\tilde{\theta}_i; w_{i,r}^t, \mathcal{D}_i)\|^2 \leq \nu, \tag{8}$$

with the number of $\nabla \tilde{h}_i$ computations $K := \mathcal{O}\big(\kappa \log\big(\frac{d}{\nu}\big)\big)$ (resp. $\mathcal{O}\big(\sqrt{\kappa} \log\big(\frac{d}{\nu}\big)\big)$) [50], where $d$ is the diameter of the search space, $\nu$ is an accuracy level, and $\mathcal{O}(\cdot)$ hides constants. The computation complexity of each client in pFedMe is $K$ times that in FedAvg. In the following lemma, we show how $\delta$ can be adjusted by controlling the (i) sampling noise using mini-batch size $|\mathcal{D}|$ and (ii) accuracy level $\nu$.

**Lemma 1.** *Let $\tilde{\theta}_i(w_{i,r}^t)$ be a solution to (8), we have*

$$\mathbb{E}\left[\|\tilde{\theta}_i(w_{i,r}^t) - \hat{\theta}_i(w_{i,r}^t)\|^2\right] \leq \delta^2 := \begin{cases} \frac{2}{(\lambda+\mu)^2}\left(\frac{\gamma_f^2}{|\mathcal{D}|} + \nu\right), & \textit{if Assumption 1(a) holds;} \\ \frac{2}{(\lambda-L)^2}\left(\frac{\gamma_f^2}{|\mathcal{D}|} + \nu\right), & \textit{if Assumption 1(b) holds, and } \lambda > L. \end{cases}$$

## 4 pFedMe: Convergence Analysis

In this section, we present the convergence of pFedMe. We first prove an intermediate result.

    **Lemma 2.** *Recall the definition of the Moreau envelope $F_i$ in pFedMe, we have*

*(a) Let Assumption 1(a) hold, then we have*

$$\frac{1}{N}\sum_{i=1}^N \|\nabla F_i(w) - \nabla F(w)\|^2 \leq 4L_F(F(w) - F(w^*)) + 2\underbrace{\frac{1}{N}\sum_{i=1}^N \|\nabla F_i(w^*)\|^2}_{=:\sigma_{F,1}^2}.$$

*(b) If Assumption 1(b) holds. Furthermore, if $\lambda > 2\sqrt{2}L$:*

$$\frac{1}{N}\sum_{i=1}^{N}\|\nabla F_i(w) - \nabla F(w)\|^2 \leq \frac{8L^2}{\lambda^2 - 8L^2}\|\nabla F(w)\|^2 + 2\underbrace{\frac{\lambda^2}{\lambda^2 - 8L^2}\sigma_f^2}_{=:\sigma_{F,2}^2}.$$

This lemma provides the bounded diversity of $F_i$, characterized by the variances $\sigma_{F,1}^2$ and $\sigma_{F,2}^2$, for strongly convex and nonconvex smooth $f_i$, respectively. While $\sigma_{F,2}^2$ is related to $\sigma_f^2$ that needs to be bounded in Assumption 3, $\sigma_{F,1}^2$ is measured only at the unique solution $w^*$ to pFedMe (for strongly convex $F_i$, $w^*$ always exists), and thus $\sigma_{F,1}^2$ is finite. These bounds are tight in the sense that $\sigma_{F,1}^2 = \sigma_{F,2}^2 = 0$ when data distribution of clients are i.i.d.

**Theorem 1** (Strongly convex pFedMe's convergence). *Let Assumptions 1(a) and 2 hold. If $T \geq \frac{2}{\hat{\eta}_1\mu_F}$, there exists an $\eta \leq \frac{\hat{\eta}_1}{\beta R}$, where $\hat{\eta}_1 := \frac{1}{6L_F(3+128\kappa_F/\beta)}$ with $\beta \geq 1$, such that*

$(a)\; \mathbb{E}\left[F(\bar{w}^T) - F(w^*)\right] \leq \mathcal{O}\big(\mathbb{E}\left[F(\bar{w}^T) - F(w^*)\right]\big) :=$

$$\mathcal{O}\Big(\Delta_0\mu_F e^{-\hat{\eta}_1\mu_F T/2}\Big) + \tilde{\mathcal{O}}\left(\frac{(N/S-1)\sigma_{F,1}^2}{\mu_F T N}\right) + \tilde{\mathcal{O}}\left(\frac{(R\sigma_{F,1}^2 + \delta^2\lambda^2)\kappa_F}{R(T\beta\mu_F)^2}\right) + \mathcal{O}\left(\frac{\lambda^2\delta^2}{\mu_F}\right)$$

$(b)\; \frac{1}{N}\sum_{i=1}^{N}\mathbb{E}\left[\|\tilde{\theta}_i^T(w_T) - w^*\|^2\right] \leq \frac{1}{\mu_F}\mathcal{O}\big(\mathbb{E}\left[F(\bar{w}_T) - F^*\right]\big) + \mathcal{O}\left(\frac{\sigma_{F,1}^2}{\lambda^2} + \delta^2\right),$

*where $\Delta_0 := \|w_0 - w^*\|^2$, $\kappa_F := \frac{L_F}{\mu_F}$, $\bar{w}_T := \sum_{t=0}^{T-1}\alpha_t w_t/A_T$ with $\alpha_t := (1-\eta\mu_F/2)^{-(t+1)}$ and $A_T := \sum_{t=0}^{T-1}\alpha_t$, and $\tilde{\mathcal{O}}(\cdot)$ hides both constants and polylogarithmic factors.*

**Corollary 1.** *When there is no client sampling (i.e., $S = N$), we can choose either (i) $\beta = \Theta(\sqrt{N}/T)$ if $\sqrt{N} \geq T$ (i.e., massive clients) or (ii) $\beta = \Theta(N\sqrt{R})$ otherwise, to obtain either linear speedup $\mathcal{O}\big(1/(TRN)\big)$ or quadratic speedup $\mathcal{O}\big(1/(TRN)^2\big)$ w.r.t computation rounds, respectively.*

**Remark 1.** Theorem 1 (a) shows the convergence of the global model w.r.t four error terms, where the expectation is w.r.t the randomness of mini-batch and client samplings. While the first term shows that a carefully chosen constant step size can reduce the initial error $\|w_0 - w^*\|^2$ linearly, the last term means that pFedMe converges towards a $\frac{\lambda^2\delta^2}{\mu_F}$-neighborhood of $w^*$, due to the approximation error $\delta$ at each local round. The second error term is due to the client sampling, which obviously is 0 when $S = N$. If we choose $S$ such that $S/N$ corresponds to a fixed ratio, e.g., 0.5, then we can obtain a linear speedup $\mathcal{O}\big(1/(TN)\big)$ w.r.t communication rounds for client sampling error. The third error term is due to client drift with multiple local updates. According to Corollary 1, we are able to obtain the quadratic speedup, while most of existing FL convergence analysis of strongly convex loss functions can only achieve linear speedup [3, 6, 45]. Theorem 1 (b) shows the convergence of personalized models in average to a ball of center $w^*$ and radius $\mathcal{O}\big(\frac{\lambda^2\delta^2}{\mu_F} + \frac{\sigma_{F,1}^2}{\lambda^2} + \delta^2\big)$, which shows that $\lambda$ can be controlled to trade off reducing the errors between $\delta^2$ and $\sigma_{F,1}^2$.

**Theorem 2** (Nonconvex and smooth pFedMe's convergence). *Let Assumptions 1(b), 2, and 3 hold. If $\eta \leq \frac{\hat{\eta}_2}{\beta R}$, where $\hat{\eta}_2 := \frac{1}{75L_F\lambda^2}$ with $\lambda \geq \sqrt{8L^2+1}$ and $\beta \geq 1$, then we have*

$(a)\; \mathbb{E}\left[\|\nabla F(w_{t^*})\|^2\right] \leq \mathcal{O}\Big(\mathbb{E}\left[\|\nabla F(w_{t^*})\|^2\right]\Big) :=$

$$\mathcal{O}\left(\frac{\Delta_F}{\hat{\eta}_2 T} + \frac{\big(\Delta_F L_F\sigma_{F,2}^2(N/S-1)\big)^{\frac{1}{2}}}{\sqrt{TN}} + \frac{(\Delta_F)^{\frac{2}{3}}\big(R\sigma_{F,2}^2 + \lambda^2\delta^2\big)^{\frac{1}{3}}}{\beta^{\frac{4}{3}}R^{\frac{1}{3}}T^{\frac{2}{3}}} + \lambda^2\delta^2\right)$$

$(b)\; \frac{1}{N}\sum_{i=1}^{N}\mathbb{E}\left[\|\tilde{\theta}_i^{t^*}(w_{t^*}) - w_{t^*}\|^2\right] \leq \mathcal{O}\Big(\mathbb{E}\left[\|\nabla F(w_{t^*})\|^2\right]\Big) + \mathcal{O}\left(\frac{\sigma_{F,2}^2}{\lambda^2} + \delta^2\right),$

*where $\Delta_F := F(w_0) - F^*$, and $t^* \in \{0,\dots,T-1\}$ is sampled uniformly.*

**Corollary 2.** *When there is no client sampling, we can choose $\beta = \Theta(N^{1/2}R^{1/4})$ and $\Theta(T^{1/3}) = \Theta((NR)^{2/3})$ to obtain a sublinear speed-up of $\mathcal{O}\big(1/(TRN)^{2/3}\big)$.*

**Remark 2.** Theorem 2 shows a similar convergence structure to that of Theorem 1, but with a sublinear rate for nonconvex case. According to Corollary 2, we are able to obtain the sublinear speedup $\mathcal{O}\big(1/(TRN)^{2/3}\big)$, while most of existing convergence analysis for nonconvex FL can only achieve a sublinear speed-up of $\mathcal{O}\big(1/\sqrt{TRN}\big)$ [3, 6, 49].

## 5 Experimental Results and Discussion

In this section, we validate the performance of `pFedMe` when the data distributions are heterogeneous and non-i.i.d. We first observe the effect of hyperparameters $K$ and $\beta$ on the convergence of `pFedMe`. We then compare `pFedMe` with FedAvg and Per-FedAvg in both $\mu$-strongly convex and nonconvex settings. The effect of other hyperparameters is presented in the supplementary document.

### 5.1 Experimental Settings

We consider a classification problem using both real (MNIST) and synthetic datasets. MNIST [51] is a handwritten digit dataset containing 10 labels and 70,000 instances. Due to the limitation on MNIST's data size, we distribute the complete dataset to $N = 20$ clients. To model a heterogeneous setting in terms of local data sizes and classes, each client is allocated a different local data size in the range of $[1165, 3834]$, and only has 2 of the 10 labels. For synthetic data, we adopt the data generation and distribution procedure from [15], using two parameters $\bar{\alpha}$ and $\bar{\beta}$ to control how much the local model and the dataset of each client differ, respectively. Specifically, the dataset serves a 10-class classifier using 60-dimensional real-valued data. We generate a synthetic dataset with $\bar{\alpha} = 0.5$ and $\bar{\beta} = 0.5$. Each client's data size is in the range of $[250, 25810]$. Finally, we distribute the data to $N = 100$ clients according to the power law in [15].

We fix the subset of clients $S = 5$ for MNIST, and $S = 10$ for Synthetic. We compare the algorithms using both cases of the same and fine-tuned learning rates, batch sizes, and number of local and global iterations. For $\mu$-strongly convex setting, we consider a $l_2$-regularized multinomial logistic regression model (MLR) with the softmax activation and cross-entropy loss functions. For nonconvex case, a two-layer deep neural network (DNN) is implemented with hidden layer of size 100 for MNIST and 20 for Synthetic using ReLU activation and a softmax layer at the end. In `pFedMe`, we use gradient descent to obtain $\delta$-approximate $\tilde{\theta}_i(w_{i,r}^t)$ and the personalized model is evaluated on the personalized parameter $\tilde{\theta}_i$ while the global model is evaluated on $w$. For the comparison with Per-FedAvg, we use its personalized model which is the local model after taking an SGD step from the global model. All datasets are split randomly with 75% and 25% for training and testing, respectively. All experiments were conducted using PyTorch [52] version 1.4.0. The code and datasets are available online[1].

### 5.2 Effect of hyperparameters

To understand how different hyperparameters such as $K$ and $\beta$ affect the convergence of `pFedMe` in both $\mu$-strongly convex and nonconvex settings, we conduct various experiments on MNIST dataset with $\eta = 0.005$ and $S = 5$. The explanations for choosing hyperparameters $R$, $|\mathcal{D}|$, and $\lambda$ are provided in the supplementary files.

**Effects of computation complexity** $K$: As $K$ allows for approximately finding the personalized model $\theta$, $K$ is also considered as a hyper-parameter of `pFedMe`. In Fig. 1, only the value of $K$ is changed during the experiments. We observe that `pFedMe` requires a small value of $K$ (around 3 to 5 steps) to approximately compute the personalized model. Larger values of $K$, such as 7, do not show the improvement on the convergence of the personalized model nor the global model. Similar to $R$, larger $K$ also requires more user's computation, which has negative effects on user energy consumption. Therefore, the value of $K = 5$ is chosen for the remaining experiments.

**Effects of** $\beta$: Fig. 2 illustrates how $\beta$ ($\beta \geq 1$) affects both the personalized and global models. It is noted that when $\beta = 1$, it is similar to model averaging of FedAvg. According to the figure, it is beneficial to shift the value of $\beta$ to be larger as it allows `pFedMe` to converge faster, especially the global model. However, turning $\beta$ carefully is also significant to prevent the divergence and instability of `pFedMe`. For example, when $\beta$ moves to the large value, to stabilize the global model as well as

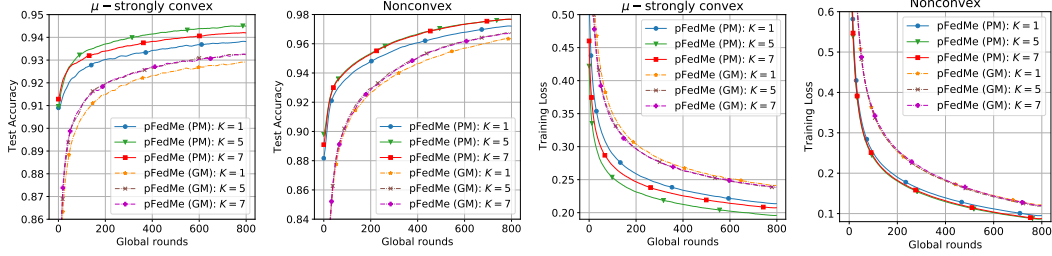

Figure 1: Effect of $K$ on the convergence of `pFedMe` in $\mu$-strongly convex and nonconvex settings on MNIST ($|\mathcal{D}| = 20$, $\lambda = 15$, $R = 20$, $\beta = 1$).

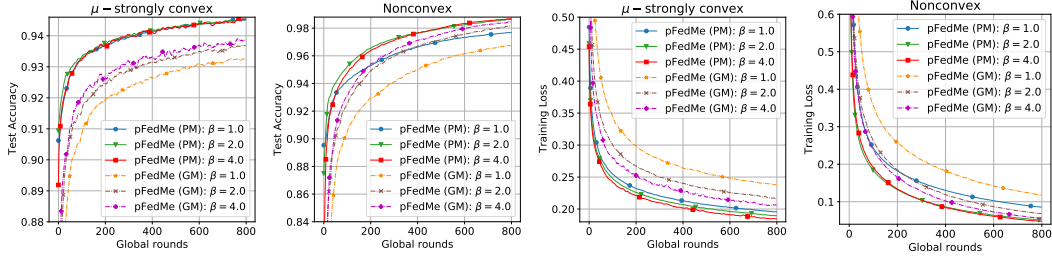

Figure 2: Effect of $\beta$ on the convergence of `pFedMe` in $\mu$-strongly convex and nonconvex settings on MNIST ($|\mathcal{D}| = 20$, $\lambda = 15$, $R = 20$, $K = 5$).

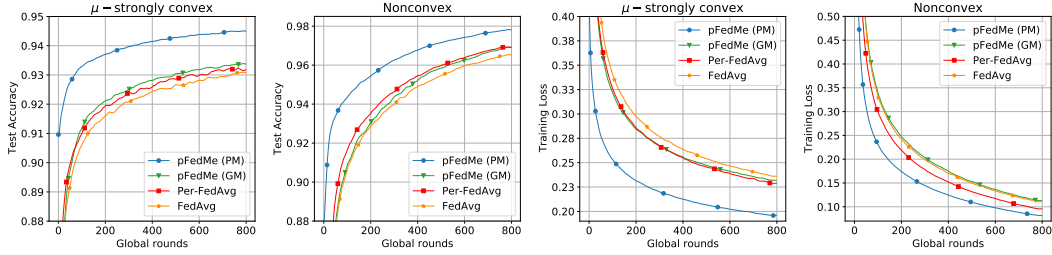

Figure 3: Performance comparison of `pFedMe`, FedAvg, and Per-FedAvg in $\mu$-strongly convex and nonconvex settings using MNIST ($\eta = 0.005$, $|\mathcal{D}| = 20$, $S = 5$, $\beta = 1$ for all experiments).

the personalized model, the smaller value of $\eta$ needs to be considered. Alternatively, $\beta$ and $\eta$ should be adjusted in inverse proportion to reach the stability of `pFedMe`.

### 5.3 Performance Comparison

In order to highlight the empirical performance of `pFedMe`, we perform several comparisons between `pFedMe`, FedAvg, and Per-FedAvg. We first use the same parameters for all algorithms as an initial comparison. As algorithms behave differently when hyperparameters are changed, we conduct a grid search on a wide range of hyperparameters to figure out the combination of fine-tuned parameters that achieves the highest test accuracy w.r.t. each algorithm. We use both personalized model (PM) and the global model (GM) of `pFedMe` for comparisons.

The comparisons for MNIST dataset are shown in Fig. 3 (the same hyperparameters) and Table. 1 (fine-tuned hyperparameters). Fig. 3 shows that the `pFedMe`'s personalized models in strongly convex setting are 1.1%, 1.3%, and 1.5% more accurate than its global model, Per-FedAvg, and FedAvg, respectively. The corresponding figures for nonconvex setting are 0.9%, 0.9%, and 1.3%. Table. 1 shows that when using fine-tuned hyperparameters, the `pFedMe`'s personalized model is the best performer in all settings.

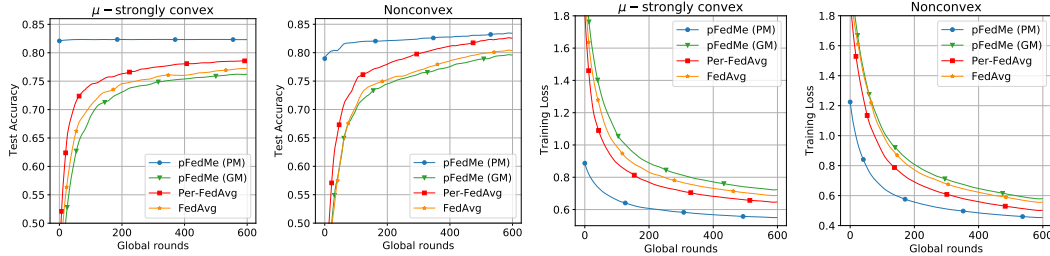

Figure 4: Performance comparison of pFedMe, FedAvg, and Per-FedAvg in $\mu$-strongly convex and nonconvex settings using Synthetic ($\eta = 0.005$, $|\mathcal{D}| = 20$, $S = 10$, $\beta = 1$ for all experiments).

Table 1: Comparison using fine-tuned hyperparameters. We fix $|\mathcal{D}| = 20$, $R = 20$, $K = 5$, and $T = 800$ for MNIST, and $T = 600$ for Synthetic, $\beta = 2$ for pFedMe ($\hat{\alpha}$ and $\hat{\beta}$ are learning rates of Per-FedAvg).

| Algorithm | Model | MNIST | | | Synthetic | | |
|---|---|---|---|---|---|---|---|
| | | $\lambda$ | $\eta\,(\hat{\alpha}, \hat{\beta})$ | Accuracy (%) | $\lambda$ | $\eta\,(\hat{\alpha}, \hat{\beta})$ | Accuracy (%) |
| FedAvg | MLR | | 0.02 | $93.96 \pm 0.02$ | | 0.02 | $77.62 \pm 0.11$ |
| Per-FedAvg | MLR | | 0.03, 0.003 | $94.37 \pm 0.04$ | | 0.02, 0.002 | $81.49 \pm 0.09$ |
| pFedMe-GM | MLR | 15 | 0.01 | $94.18 \pm 0.06$ | 20 | 0.01 | $78.65 \pm 0.25$ |
| pFedMe-PM | MLR | 15 | 0.01 | $\mathbf{95.62} \pm 0.04$ | 20 | 0.01 | $\mathbf{83.20} \pm 0.06$ |
| FedAvg | DNN | | 0.02 | $98.79 \pm 0.03$ | | 0.03 | $83.64 \pm 0.22$ |
| Per-FedAvg | DNN | | 0.02, 0.001 | $98.90 \pm 0.02$ | | 0.01, 0.001 | $85.01 \pm 0.10$ |
| pFedMe-GM | DNN | 30 | 0.01 | $99.16 \pm 0.03$ | 30 | 0.01 | $84.17 \pm 0.35$ |
| pFedMe-PM | DNN | 30 | 0.01 | $\mathbf{99.46} \pm 0.01$ | 30 | 0.01 | $\mathbf{86.36} \pm 0.15$ |

For Synthetic dataset, the comparisons for utilizing the same parameters and the fine-tuned parameter are presented in Fig. 4 and Table. 1, respectively. In Fig. 4, although pFedMe's global model performs worse than others on test accuracy and training loss, the personalized model shows its advantages by achieving the highest test figures. Fig. 4 shows that pFedMe's personalized model is 6.1%, 3.8%, and 5.2% more accurate than its global model, Per-FedAvg, and FedAvg, respectively. The respective figures for the nonconvex setting are 3.9%, 0.7%, and 3.1%. In addition, with fine-tuned hyperparameters in Table. 1, the personalized model of pFedMe outperforms others in all settings while the global model of pFedMe only performs better than FedAvg.

From the experimental results, when the data among clients are non-i.i.d, both pFedMe and Per-Avg gain higher testing accuracy than FedAvg as they allow the global model to be personalized for a specific client. However, by optimizing the personalized model approximately with multiple gradient updates and avoiding computing the Hessian matrix, the personalized model of pFedMe is more advantageous than Per-FedAvg in terms of the convergence rate and the computation complexity.

# 6 Conclusion

In this paper, we propose pFedMe as a personalized FL algorithm that can adapt to the statistical diversity issue to improve the FL performance. Our approach makes use of the Moreau envelope function which helps decompose the personalized model optimization from global model learning, which allows pFedMe to update the global model similarly to FedAvg, yet in parallel to optimize the personalized model w.r.t each client's local data distribution. Theoretical results show that pFedMe can achieve the state-of-the-art convergence speedup rate. Experimental results demonstrate that pFedMe outperforms the vanilla FedAvg and the meta-learning based personalized FL algorithm Per-FedAvg in both convex and non-convex settings, using both real and synthetic datasets. Finally, the degree to which personalization becomes provably useful is a topic of experimental research, as parameters will need to be adapted to each dataset and federated setting.

## Broader Impact

There have been numerous applications of FL in practice. One notable commercial FL usage, which has proved successful in recent years, is in the next-character prediction task on mobile devices. However, we believe this technology promises many more breakthroughs in a number of fields in the near future with the help of personalized FL models. In health care, for example, common causes of a disease can be identified from many patients without the need to have access to their raw data. The development of capable personalized models helps build better predictors on patients' conditions, allowing for faster, more efficient diagnosis and treatment.

As much as FL promises, it also comes with a number of challenges. First, an important societal requirement when deploying such technique is that the server must explain which clients' data will be participated and which will not. The explainability and interpretability of a system are necessary for the sake of public understanding and making informed consent. Second, to successfully preserve privacy, FL has to overcome malicious actors who possibly interfere in the training process during communication. The malicious behaviors include stealing personalized models from the server, performing adversarial attacks such as changing a personalized model on some examples while remaining a good performance on average, and attempting to alter a model. Finally, an effective and unbiased FL system must be aware that data and computational power among clients can be extremely uneven in practice and, therefore, must ensure that the contribution of each client to the global model is adjusted to its level of distribution. These challenges help necessitate future research in decentralized learning in general and personalized FL in particular.

## Acknowledgments and Disclosure of Funding

This research is funded by Vietnam National University HoChiMinh City (VNU-HCM) under grant number DS2020-28-01. Tuan Dung Nguyen's work is supported by the Faculty of Engineering Scholarship at the University of Sydney.

## Footnotes

[1]https://github.com/CharlieDinh/pFedMe

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
