[Supplementary Material]

# Personalized Federated Learning with Moreau Envelopes: Supplementary Materials

**Canh T. Dinh**[1], **Nguyen H. Tran**[1], **Tuan Dung Nguyen**[1,2]

[1]The University of Sydney, Australia
`tdin6081@uni.sydney.edu.au, nguyen.tran@sydney.edu.au`
[2]The University of Melbourne, Australia
`tuandungn@unimelb.edu.au`

## Abstract

In this appendix we provide proofs for the theorems and lemmas in the paper "Personalized Federated Learning with Moreau Envelopes", as well as additional experimental settings and results.

## A  Proof of the Results

In this section, we first provide some existing results useful for following proofs. We then present the proofs of Lemma 1, Lemma 2, Theorem 1, and Theorem 2.

### A.1  Review of useful existing results

**Proposition 2.** *[1, Theorems 2.1.5 and 2.1.10] If a function $F_i(\cdot)$ is $L_F$-smooth and $\mu_F$-strongly convex, $\forall w, w'$, we have the following useful inequalities, in respective order,*

$$\|\nabla F_i(w) - \nabla F_i(w')\|^2 \leq 2L_F(F_i(w) - F_i(w') - \langle \nabla F_i(w'), w - w' \rangle)$$

$$\mu_F \|w - w'\| \leq \|\nabla F_i(w) - \nabla F_i(w')\|.$$

*where $w^*$ is the solution to problem $\min_{w \in \mathbb{R}^d} F_i(w)$, i.e., $\nabla F_i(w^*) = 0$.*

**Proposition 3.** *For any vector $x_i \in \mathbb{R}^d$, $i = 1, \ldots, M$, by Jensen's inequality, we have*

$$\left\| \sum_{i=1}^{M} x_i \right\|^2 \leq M \sum_{i=1}^{M} \|x_i\|^2.$$

### A.2  Proof of Lemma 1

*Proof.* We first prove case (a). Let $h_i(\theta_i; w_{i,r}^t) := f_i(\theta_i) + \frac{\lambda}{2}\|\theta_i - w_{i,r}^t\|^2$. Then $h_i(\theta_i; w_{i,r}^t)$ is $(\lambda + \mu)$-strongly convex with its unique solution $\hat{\theta}_i(w_{i,r}^t)$. Then, by Proposition 2, we have

$$
\begin{aligned}
\|\tilde{\theta}_i(w_{i,r}^t) - \hat{\theta}_i(w_{i,r}^t)\|^2 &\leq \frac{1}{(\lambda + \mu)^2} \|\nabla h_i(\tilde{\theta}_i; w_{i,r}^t)\|^2 \\
&\leq \frac{2}{(\lambda + \mu)^2} \Big( \|\nabla h_i(\tilde{\theta}_i; w_{i,r}^t) - \nabla \tilde{h}_i(\tilde{\theta}_i; w_{i,r}^t, \mathcal{D}_i)\|^2 + \|\nabla \tilde{h}_i(\tilde{\theta}_i; w_{i,r}^t, \mathcal{D}_i)\|^2 \Big) \\
&\leq \frac{2}{(\lambda + \mu)^2} \Big( \|\nabla \tilde{f}_i(\tilde{\theta}_i; \mathcal{D}_i) - \nabla f_i(\tilde{\theta}_i)\|^2 + \nu \Big) \\
&= \frac{2}{(\lambda + \mu)^2} \Big( \frac{1}{|\mathcal{D}|^2} \Big\| \sum_{\xi_i \in \mathcal{D}_i} \nabla \tilde{f}_i(\tilde{\theta}_i; \xi_i) - \nabla f_i(\tilde{\theta}_i) \Big\|^2 + \nu \Big),
\end{aligned}
$$

where the second inequality is by Proposition 3. Taking expectation to both sides, we have

$$\mathbb{E}\left[\|\tilde{\theta}_i(w_{i,r}^t) - \hat{\theta}_i(w_{i,r}^t)\|^2\right] = \frac{2}{(\lambda+\mu)^2}\left(\frac{1}{|\mathcal{D}|^2}\sum_{\xi_i\in\mathcal{D}_i}\mathbb{E}_{\xi_i}\left[\|\nabla\tilde{f}_i(\tilde{\theta}_i;\xi_i) - \nabla f_i(\tilde{\theta}_i)\|^2\right] + \nu\right)$$

$$\leq \frac{2}{(\lambda+\mu)^2}\left(\frac{\gamma_f^2}{|\mathcal{D}|} + \nu\right),$$

where the first equality is due to $\mathbb{E}\left[\|\sum_{i=1}^M X_i - \mathbb{E}\left[X_i\right]\|^2\right] = \sum_{i=1}^M \mathbb{E}\left[\|X_i - \mathbb{E}\left[X_i\right]\|\right]^2$ with $M$ independent random variables $X_i$ and the unbiased estimate $\mathbb{E}\left[\nabla\tilde{f}_i(\tilde{\theta}_i;\xi_i)\right] = \nabla f_i(\tilde{\theta}_i)$, and the last inequality is due to Assumption 2.

The proof of case (b) follows similarly, considering that $h_i(\theta_i; w_{i,r}^t)$ is $(\lambda - L)$-strongly convex. $\quad\square$

### A.3 Proof of Lemma 2

*Proof.* We first prove case (a).

$$\frac{1}{N}\sum_{i=1}^N\|\nabla F_i(w)\|^2 \leq \frac{1}{N}\sum_{i=1}^N 2\Big(\|\nabla F_i(w) - \nabla F_i(w^*)\|^2 + \|\nabla F_i(w^*)\|^2\Big)$$

$$\leq 4L_F(F(w) - F(w^*)) + \frac{2}{N}\sum_{i=1}^N\|\nabla F_i(w^*)\|^2,$$

where the first and the second inequalities are due to Propositions 3 and 2, respectively.

We next prove case (b):

$$\|\nabla F_i(w) - \nabla F(w)\|^2$$

$$= \left\|\lambda\big(w - \hat{\theta}_i(w)\big) - \frac{1}{N}\sum_{j=1}^N \lambda\big(w - \hat{\theta}_j(w)\big)\right\|^2$$

$$= \left\|\nabla f_i(\hat{\theta}_i(w)) - \frac{1}{N}\sum_{j=1}^N \nabla f_j(\hat{\theta}_j(w))\right\|^2$$

$$= 2\left\|\nabla f_i(\hat{\theta}_i(w)) - \frac{1}{N}\sum_{j=1}^N \nabla f_j(\hat{\theta}_i(w))\right\|^2 + 2\left\|\frac{1}{N}\sum_{j=1}^N \nabla f_j(\hat{\theta}_i(w)) - \nabla f_j(\hat{\theta}_j(w))\right\|^2,$$

where the second inequality is due to the first-order condition $\nabla f_i(\hat{\theta}_i(w)) - \lambda\big(w - \hat{\theta}_i(w)\big) = 0$, and the last one is due to Proposition 3. Taking the average over the number of clients, we have

$$\frac{1}{N}\sum_{i=1}^N\|\nabla F_i(w) - \nabla F(w)\|^2 \leq 2\sigma_f^2 + \frac{2}{N^2}\sum_{i=1}^N\sum_{j=1}^N\|\nabla f_j(\hat{\theta}_i(w)) - \nabla f_j(\hat{\theta}_j(w))\|^2 \qquad (9)$$

$$\leq 2\sigma_f^2 + \frac{2L^2}{N^2}\sum_{i=1}^N\sum_{j=1}^N\|\hat{\theta}_i(w) - \hat{\theta}_j(w)\|^2 \qquad (10)$$

$$\leq 2\sigma_f^2 + \frac{2L^2}{N^2}\sum_{i=1}^N\sum_{j=1}^N 2\Big(\|\hat{\theta}_i(w) - w\|^2 + \|\hat{\theta}_j(w) - w\|^2\Big) \qquad (11)$$

$$\leq 2\sigma_f^2 + \frac{2L^2}{N^2}\sum_{i=1}^N\sum_{j=1}^N\frac{2}{\lambda^2}\Big(\|\nabla F_i(w)\|^2 + \|\nabla F_j(w)\|^2\Big) \qquad (12)$$

$$= 2\sigma_f^2 + \frac{8L^2}{\lambda^2}\frac{1}{N}\sum_{i=1}^N\|\nabla F_i(w)\|^2$$

$$= 2\sigma_f^2 + \frac{8L^2}{\lambda^2}\left[\frac{1}{N}\sum_{i=1}^N\|\nabla F_i(w) - \nabla F(w)\|^2 + \|\nabla F(w)\|^2\right] \qquad (13)$$

where (9) is due to Assumption 3 and Proposition 3, which is also used for (11), (10) is due to $L$-smoothness of $f_i(\cdot)$, (12) is due to Proposition 1, (13) is by the fact that $\mathbb{E}\big[\|X\|^2\big] = \mathbb{E}\big[\|X - \mathbb{E}[X]\|^2\big] + \mathbb{E}[\|X\|]^2$ for any vector of random variable $X$. Finally, by re-arranging the terms of (13), we obtain

$$\frac{1}{N}\sum_{i=1}^{N}\big\|\nabla F_i(w) - \nabla F(w)\big\|^2 \le \frac{2\lambda^2}{\lambda^2 - 8L^2}\sigma_f^2 + \frac{8L^2}{\lambda^2 - 8L^2}\big\|\nabla F(w)\big\|^2.$$

□

### A.4 Proof of Theorem 1

We first define additional notations for the ease of analysis. We next provide supporting lemmas, and finally we will combine them to complete the proof of Theorem 1.

#### A.4.1 Additional notations

We re-write the local update as follows

$$w_{i,r+1}^t = w_{i,r}^t - \eta \underbrace{\lambda(w_{i,r}^t - \tilde{\theta}_i(w_{i,r}^t))}_{=:\, g_{i,r}^t}$$

which implies

$$\eta\sum_{r=0}^{R-1} g_{i,r}^t = \sum_{r=0}^{R-1}\big(w_{i,r}^t - w_{i,r+1}^t\big) = w_{i,0}^t - w_{i,R}^t = w_t - w_{i,R}^t,$$

where $g_{i,r}^t$ can be considered as the biased estimate of $\nabla F_i(w_{i,r}^t)$ since $\mathbb{E}\big[g_{i,r}^t\big] \ne \nabla F_i(w_{i,r}^t)$. We also re-write the global update as follows

$$\begin{aligned}
w_{t+1} &= (1-\beta)w_t + \frac{\beta}{S}\sum_{i\in\mathcal{S}^t} w_{i,R}^t \\
&= w_t - \frac{\beta}{S}\sum_{i\in\mathcal{S}^t}(w_t - w_{i,R}^t) \\
&= w_t - \underbrace{\eta\beta R}_{=:\,\tilde{\eta}}\underbrace{\frac{1}{SR}\sum_{i\in\mathcal{S}^t}\sum_{r=0}^{R-1} g_{i,r}^t}_{=:\,g_t},
\end{aligned}$$

where $\tilde{\eta}$ and $g_t$ can be interpreted as the step size and approximate stochastic gradient, respectively, of the global update.

#### A.4.2 Supporting lemmas

**Lemma 3** (One-step global update). *Let Assumption 1(b) hold. We have*

$$\mathbb{E}\big[\|w_{t+1} - w^*\|^2\big] \le \Big(1 - \frac{\tilde{\eta}\mu_F}{2}\Big)\mathbb{E}\big[\|w_t - w^*\|^2\big] - \tilde{\eta}\big(2 - 6L_F\tilde{\eta}\big)\mathbb{E}\big[F(w_t) - F(w^*)\big]$$

$$+ \frac{\tilde{\eta}(3\tilde{\eta} + 2/\mu_F)}{NR}\sum_{i,r}^{N,R}\mathbb{E}\Big[\big\|g_{i,r} - \nabla F_i(w_t)\big\|^2\Big] + 3\tilde{\eta}^2\mathbb{E}\bigg[\Big\|\frac{1}{S}\sum_{i\in\mathcal{S}^t}\nabla F_i(w_t) - \nabla F(w_t)\Big\|^2\bigg],$$

*where $\sum_{i,r}^{N,R}$ is used as an alternative for $\sum_{i=1}^{N}\sum_{r=0}^{R-1}$.*

*Proof.* Denote the expectation conditioning on all randomness prior to round $t$ by $\mathbb{E}_t$. We have

$$\begin{aligned}
\mathbb{E}_t\big[\|w_{t+1} - w^*\|^2\big] &= \mathbb{E}_t\big[\|w_t - \tilde{\eta}g_t - w^*\|^2\big] \\
&= \|w_t - w^*\|^2 - 2\tilde{\eta}\,\mathbb{E}_t\big[\langle g_t, w_t - w^*\rangle\big] + \tilde{\eta}^2\mathbb{E}_t\big[\|g_t\|^2\big]. \quad (14)
\end{aligned}$$

We first take expectation of the second term of (14) w.r.t client sampling

$$\begin{aligned}
-\mathbb{E}_{\mathcal{S}_t}\big[\langle g_t, w_t - w^*\rangle\big] &= -\langle\mathbb{E}_{\mathcal{S}_t}[g_t], w_t - w^*\rangle \\
&= -\frac{1}{NR}\sum_{i,r}^{N,R}\Big(\langle g_{i,r}^t - \nabla F_i(w_t), w_t - w^*\rangle + \langle\nabla F_i(w_t), w_t - w^*\rangle\Big), \quad (15)
\end{aligned}$$

where the second equality is obtained by having $\mathbb{E}_{\mathcal{S}_t}[g_t] = \mathbb{E}_{\mathcal{S}_t}\big[\frac{1}{SR}\sum_{i,r}^{\mathcal{S}^t,R} g_{i,r}^t\big] = \frac{1}{SR}\sum_{i,r}^{N,R} g_{i,r}^t\mathbb{E}_{\mathcal{S}_t}\big[\mathbb{I}_{i\in S_t}\big] = \frac{1}{NR}\sum_{i,r}^{N,R} g_{i,r}^t$, where $\mathbb{I}_A$ is the indicator function of an event $A$ and thus $\mathbb{E}_{\mathcal{S}_t}\big[\mathbb{I}_{i\in S_t}\big] = S/N$ due to uniform sampling. We then bound two terms of (15) as follows

$$-\frac{1}{N}\sum_{i=1}^{N}\langle\nabla F_i(w_t), w_t - w^*\rangle \le F(w^*) - F(w_t) - \frac{\mu_F}{2}\|w_t - w^*\|^2 \tag{16}$$

$$-\frac{2}{NR}\sum_{i,r}^{N,R}\langle g_{i,r}^t - \nabla F_i(w_t), w_t - w^*\rangle \le \frac{1}{NR}\sum_{i,r}^{N,R}\left(\frac{2}{\mu_F}\|g_{i,r}^t - \nabla F_i(w_t)\|^2 + \frac{\mu_F}{2}\|w_t - w^*\|^2\right) \tag{17}$$

where the first and second inequalities are due to $\mu_F$-strongly convex $F_i(\cdot)$ and the Peter Paul inequality, respectively.

We next take expectation of the last term of (14) w.r.t client sampling

$$\mathbb{E}_{\mathcal{S}_t}\left[\|g_t\|^2\right] = \mathbb{E}_{\mathcal{S}_t}\left\|\frac{1}{SR}\sum_{i,r}^{\mathcal{S}^t,R} g_{i,r}^t\right\|^2$$

$$\le 3\mathbb{E}_{\mathcal{S}_t}\left[\left\|\frac{1}{SR}\sum_{i,r}^{\mathcal{S}^t,R} g_{i,r}^t - \nabla F_i(w_t)\right\|^2 + \left\|\frac{1}{S}\sum_{i\in\mathcal{S}^t}\nabla F_i(w_t) - \nabla F(w_t)\right\|^2 + \|\nabla F(w_t)\|^2\right]$$

$$\le \frac{3}{NR}\sum_{i,r}^{N,R}\left\|g_{i,r}^t - \nabla F_i(w_t)\right\|^2 + 3\mathbb{E}_{\mathcal{S}_t}\left\|\frac{1}{S}\sum_{i\in\mathcal{S}^t}\nabla F_i(w_t) - \nabla F(w_t)\right\|^2 + 6L_F\big(F(w_t) - F(w^*)\big), \tag{18}$$

where the first inequality is by Proposition 3, and the second inequality is by Proposition 2 and

$$\mathbb{E}_{\mathcal{S}_t}\left[\left\|\frac{1}{SR}\sum_{i,r}^{\mathcal{S}^t,R} g_{i,r}^t - \nabla F_i(w_t)\right\|^2\right] \le \frac{1}{SR}\mathbb{E}_{\mathcal{S}_t}\left[\sum_{i,r}^{\mathcal{S}^t,R}\left\|g_{i,r}^t - \nabla F_i(w_t)\right\|^2\right]$$

$$= \frac{1}{SR}\sum_{i,r}^{N,R}\left\|g_{i,r}^t - \nabla F_i(w_t)\right\|^2\mathbb{E}_{\mathcal{S}_t}\left[\mathbb{I}_{i\in S_t}\right]$$

$$= \frac{1}{NR}\sum_{i,r}^{N,R}\left\|g_{i,r}^t - \nabla F_i(w_t)\right\|^2.$$

By substituting (16), (17), and (18) into (14), and take expectation with all history, we finish the proof. $\qquad\square$

**Lemma 4** (Bounded diversity of $F_i$ w.r.t client sampling).

$$\mathbb{E}_{\mathcal{S}_t}\left\|\frac{1}{S}\sum_{i\in\mathcal{S}^t}\nabla F_i(w_t) - \nabla F(w_t)\right\|^2 \le \frac{N/S - 1}{N - 1}\sum_{i=1}^{N}\frac{1}{N}\|\nabla F_i(w_t) - \nabla F(w_t)\|^2.$$

*Proof.* We use similar proof arguments in [2, Lemma 5] as follows

$$\mathbb{E}_{\mathcal{S}_t}\left\|\frac{1}{S}\sum_{i\in\mathcal{S}^t}\nabla F_i(w_t) - \nabla F(w_t)\right\|^2 = \frac{1}{S^2}\mathbb{E}_{\mathcal{S}_t}\left\|\sum_{i=1}^{N}\mathbb{I}_{i\in S_t}\big(\nabla F_i(w_t) - \nabla F(w_t)\big)\right\|^2$$

$$= \frac{1}{S^2}\left[\sum_{i=1}^{N}\mathbb{E}_{\mathcal{S}_t}\big[\mathbb{I}_{i\in S_t}\big]\left\|\nabla F_i(w_t) - \nabla F(w_t)\right\|^2\right.$$

$$\left.+ \sum_{i\ne j}\mathbb{E}_{\mathcal{S}_t}\big[\mathbb{I}_{i\in S_t}\mathbb{I}_{j\in S_t}\big]\langle\nabla F_i(w_t) - \nabla F(w_t), \nabla F_j(w_t) - \nabla F(w_t)\rangle\right]$$

$$= \frac{1}{SN}\sum_{i=1}^{N}\|\nabla F_i(w_t) - \nabla F(w_t)\|^2 + \sum_{i\neq j}\frac{S-1}{SN(N-1)}\langle \nabla F_i(w_t) - \nabla F(w_t), \nabla F_j(w_t) - \nabla F(w_t)\rangle$$

$$= \frac{1}{SN}\left(1 - \frac{S-1}{N-1}\right)\sum_{i=1}^{N}\|\nabla F_i(w_t) - \nabla F(w_t)\|^2$$

$$= \frac{N/S - 1}{N-1}\sum_{i=1}^{N}\frac{1}{N}\|\nabla F_i(w_t) - \nabla F(w_t)\|^2,$$

where the third equality is due to $\mathbb{E}_{\mathcal{S}_t}\left[\mathbb{I}_{i\in S_t}\right] = \mathbb{P}(i\in S_t) = \frac{S}{N}$ and $\mathbb{E}_{\mathcal{S}_t}\left[\mathbb{I}_{i\in S_t}\mathbb{I}_{j\in S_t}\right] = \mathbb{P}(i,j\in S_t) = \frac{S(S-1)}{N(N-1)}$ for all $i\neq j$, and the fourth equality is by $\sum_{i=1}^{N}\|\nabla F_i(w_t) - \nabla F(w_t)\|^2 + \sum_{i\neq j}\langle \nabla F_i(w_t) - \nabla F(w_t), \nabla F_j(w_t) - \nabla F(w_t)\rangle = 0$. $\square$

**Lemma 5** (Bounded client drift error). *If $\tilde{\eta} \leq \frac{\beta}{2L_F} \Leftrightarrow \eta \leq \frac{1}{2RL_F}$, we have*

$$\frac{1}{NR}\sum_{i,r}^{N,R}\mathbb{E}\left[\|g_{i,r}^t - \nabla F_i(w_t)\|^2\right] \leq 2\lambda^2\delta^2 + \frac{16L_F^2\tilde{\eta}^2}{\beta^2}\left(3\frac{1}{N}\sum_{i=1}^{N}\mathbb{E}\left[\|\nabla F_i(w_t)\|^2\right] + \frac{2\lambda^2\delta^2}{R}\right).$$

*Proof.*
$$\mathbb{E}\left[\|g_{i,r}^t - \nabla F_i(w_t)\|^2\right] \leq 2\mathbb{E}\left[\|g_{i,r}^t - \nabla F_i(w_{i,r}^t)\|^2 + \|\nabla F_i(w_{i,r}^t) - \nabla F_i(w_t)\|^2\right]$$
$$\leq 2\left(\lambda^2\mathbb{E}\left[\|\tilde{\theta}_i(w_{i,r}^t) - \hat{\theta}_i(w_{i,r}^t)\|^2\right] + L_F^2\mathbb{E}\left[\|w_{i,r}^t - w_t\|^2\right]\right)$$
$$\leq 2\left(\lambda^2\delta^2 + L_F^2\mathbb{E}\left[\|w_{i,r}^t - w_t\|^2\right]\right), \tag{19}$$

where the first and second inequalities are due to Propositions 3 and 2, respectively. We next bound the drift of local update of client $i$ from global model $\|w_{i,r}^t - w_t\|^2$ as follows

$$\mathbb{E}\left[\|w_{i,r}^t - w_t\|^2\right] = \mathbb{E}\left[\|w_{i,r-1}^t - w_t - \eta g_{i,r-1}^t\|^2\right]$$
$$\leq 2\mathbb{E}\left[\|w_{i,r-1}^t - w_t - \eta\nabla F_i(w_t)\|^2 + \eta^2\|g_{i,r-1}^t - \nabla F_i(w_t)\|^2\right]$$
$$\leq 2\left(1 + \frac{1}{2R}\right)\mathbb{E}\left[\|w_{i,r-1}^t - w_t\|^2\right] + 2(1+2R)\eta^2\mathbb{E}\left[\|\nabla F_i(w_t)\|^2\right]$$
$$\quad + 4\eta^2\left(\lambda^2\delta^2 + L_F^2\mathbb{E}\left[\|w_{i,r-1}^t - w_t\|^2\right]\right)$$
$$= 2\left(1 + \frac{1}{2R} + 2\eta^2 L_F^2\right)\mathbb{E}\left[\|w_{i,r-1}^t - w_t\|^2\right] + 2(1+2R)\eta^2\mathbb{E}\left[\|\nabla F_i(w_t)\|^2\right] + 4\eta^2\lambda^2\delta^2$$
$$\leq 2\left(1 + \frac{1}{R}\right)\mathbb{E}\left[\|w_{i,r-1}^t - w_t\|^2\right] + 2(1+2R)\eta^2\mathbb{E}\left[\|\nabla F_i(w_t)\|^2\right] + 4\eta^2\lambda^2\delta^2 \tag{20}$$
$$\leq \left(\frac{6\tilde{\eta}^2}{\beta^2 R}\mathbb{E}\left[\|\nabla F_i(w_t)\|^2\right] + \frac{4\tilde{\eta}^2\lambda^2\delta^2}{\beta^2 R^2}\right)\sum_{r=0}^{R-1}2\left(1+\frac{1}{R}\right)^r \tag{21}$$
$$\leq \frac{8\tilde{\eta}^2}{\beta^2}\left(3\mathbb{E}\left[\|\nabla F_i(w_t)\|^2\right] + \frac{2\lambda^2\delta^2}{R}\right), \tag{22}$$

where (20) is by having $2\eta^2 L_F^2 = 2L_F^2\frac{\tilde{\eta}^2}{\beta^2 R^2} \leq \frac{1}{2R^2} \leq \frac{1}{2R}$ when $\tilde{\eta}^2 \leq \frac{\beta^2}{4L_F^2}$, for all $R \geq 1$. (21) is due to unrolling (20) recursively, and $2(1+2R)\eta^2 = 2(1+2R)\frac{\tilde{\eta}^2}{\beta^2 R^2} \leq \frac{6\tilde{\eta}^2}{\beta^2 R}$ because $\frac{1+2R}{R} \leq 3$ when $R \geq 1$. We have (22) because $\sum_{r=0}^{R-1}(1+1/R)^r = \frac{(1+1/R)^R - 1}{1/R} \leq \frac{e-1}{1/R} \leq 2R$, by using the facts that $\sum_{i=0}^{n-1}x^i = \frac{x^n-1}{x-1}$ and $(1+\frac{x}{n})^n \leq e^x$ for any $x \in \mathbb{R}, n \in \mathbb{N}$. Substituting (22) to (19), we obtain

$$\mathbb{E}\left[\|g_{i,r}^t - \nabla F_i(w_t)\|^2\right] \leq 2\lambda^2\delta^2 + \frac{16\tilde{\eta}^2 L_F^2}{\beta^2}\left(3\mathbb{E}\left[\|\nabla F_i(w_t)\|^2\right] + \frac{2\lambda^2\delta^2}{R}\right). \tag{23}$$

By taking average over $N$ and $R$, we finish the proof.

$\square$

### A.4.3 Completing the proof of Theorem 1

*Proof.* Before proving the main theorem, we derive the first auxiliary result:

$$\mathbb{E}\left[\left\|\frac{1}{S}\sum_{i\in\mathcal{S}^t}\nabla F_i(w_t)-\nabla F(w_t)\right\|^2\right] \leq \frac{N/S-1}{N-1}\sum_{i=1}^{N}\frac{1}{N}\mathbb{E}\left[\|\nabla F_i(w_t)-\nabla F(w_t)\|^2\right] \quad (24)$$

$$\leq \frac{N/S-1}{N-1}\left(4L_F\mathbb{E}\left[F(w_t)-F(w^*)\right]+2\sigma_{F,1}^2\right), \quad (25)$$

where (24) is by Lemma 4 and (25) is by Lemma 2 (a).

The second auxiliary result is as follows

$$\frac{\tilde{\eta}(3\tilde{\eta}+2/\mu_F)}{NR}\sum_{i,r}^{N,R}\mathbb{E}\left[\|g_{i,r}^t-\nabla F_i(w_t)\|^2\right]$$

$$\leq \tilde{\eta}\frac{16\delta^2\lambda^2}{\mu_F}+\frac{\tilde{\eta}^3}{\beta^2}\frac{128L_F^2}{\mu_F}\sum_{i=1}^{N}\frac{1}{N}\left(3\mathbb{E}\left[\|\nabla F_i(w_t)\|^2\right]+\frac{2\delta^2\lambda^2}{R}\right) \quad (26)$$

$$\leq \tilde{\eta}\frac{16\delta^2\lambda^2}{\mu_F}+\frac{\tilde{\eta}^3}{\beta^2}\frac{128L_F^2}{\mu_F}\sum_{i=1}^{N}\frac{1}{N}\left(6\mathbb{E}\left[\|\nabla F_i(w_t)-\nabla F_i(w^*)\|^2\right]+6\mathbb{E}\left[\|\nabla F_i(w^*)\|^2\right]+\frac{2\delta^2\lambda^2}{R}\right) \quad (27)$$

$$\leq \tilde{\eta}\frac{16\delta^2\lambda^2}{\mu_F}+\frac{\tilde{\eta}^3}{\beta^2}\frac{128L_F^2}{\mu_F}\left(12L_F\mathbb{E}\left[F(w_t)-F(w^*)\right]+\frac{2(3R\sigma_{F,1}^2+\delta^2\lambda^2)}{R}\right) \quad (28)$$

$$\leq \tilde{\eta}\frac{16\delta^2\lambda^2}{\mu_F}+\frac{\tilde{\eta}^2}{\beta}768\kappa_FL_F\mathbb{E}\left[F(w_t)-F(w^*)\right]+\frac{\tilde{\eta}^3}{\beta^2}\frac{256(3R\sigma_{F,1}^2+\delta^2\lambda^2)\kappa_F}{R}, \quad (29)$$

where we have (26) by using Lemma 5 and $3\tilde{\eta}+2/\mu_F\leq 8/\mu_F$ when $\tilde{\eta}\leq 2/\mu_F$. (27) is by the fact that $\mathbb{E}\left[\|X\|^2\right]=\mathbb{E}\left[\|X-\mathbb{E}[X]\|^2\right]+\mathbb{E}[\|X\|]^2$ for any vector of random variable $X$. (28) is due to Lemma 2 and $\|\nabla F(w_t)\|^2\leq 2L_F\left(F(w_t)-F(w^*)\right)$ by $L_F$-smoothness of $F(\cdot)$. (29) is due to $\tilde{\eta}\leq\frac{\beta}{2L_F}$ and $\kappa_F:=\frac{L_F}{\mu_F}$.

By substituting (25) and (28) into Lemma 3, we have

$$\mathbb{E}\left[\|w_{t+1}-w^*\|^2\right]\leq$$

$$\left(1-\frac{\tilde{\eta}\mu_F}{2}\right)\mathbb{E}\left[\|w_t-w^*\|^2\right]-\tilde{\eta}\overbrace{\left[2-\tilde{\eta}\,L_F\left(6+12\frac{N/S-1}{N-1}+\frac{768\kappa_F}{\beta}\right)\right]}^{\geq 1\text{ when }\tilde{\eta}\text{ satisfied (31)}}\mathbb{E}\left[F(w_t)-F(w^*)\right]$$

$$+\tilde{\eta}\underbrace{\frac{16\delta^2\lambda^2}{\mu_F}}_{=:C_1}+\tilde{\eta}^2\underbrace{\frac{6\sigma_{F,1}^2(N/S-1)}{N-1}}_{=:C_2}+\frac{\tilde{\eta}^3}{\beta^2}\underbrace{\frac{256(3R\sigma_{F,1}^2+\delta^2\lambda^2)\kappa_F}{R}}_{=:C_3}$$

$$\leq \left(1-\frac{\tilde{\eta}\mu_F}{2}\right)\mathbb{E}\left[\|w_t-w^*\|^2\right]-\tilde{\eta}\,\mathbb{E}\left[F(w_t)-F(w^*)\right]+\tilde{\eta}C_1+\tilde{\eta}^2C_2+\frac{\tilde{\eta}^3}{\beta^2}C_3, \quad (30)$$

where we have (30) by using the fact that $\frac{N/S-1}{N-1}\leq 1$ for the following inequality

$$2-\tilde{\eta}\,L_F\left(6+12\frac{N/S-1}{N-1}+\frac{768\kappa_F}{\beta}\right)\geq 2-6\tilde{\eta}\,L_F\left(3+\frac{128\kappa_F}{\beta}\right)\geq 1$$

with the condition

$$\tilde{\eta}\leq\frac{1}{6L_F(3+128\kappa_F/\beta)}=:\hat{\eta}_1. \quad (31)$$

We note that $\hat{\eta}_1\leq\min\left\{\frac{\beta}{2L_F},\frac{2}{\mu_F}\right\}$ with $\beta\geq 1$ and $L_F\geq\mu_F$.

Let $\Delta_t := \|w_t - w^*\|^2$. By re-arranging the terms and multiplying both sides of (30) with $\frac{\alpha_t}{\tilde{\eta} A_T}$, where $A_T := \sum_{t=0}^{T-1} \alpha_t$, then we have

$$\sum_{t=0}^{T-1} \frac{\alpha_t \mathbb{E}\left[F(w_t)\right]}{A_T} - F(w^*) \leq \sum_{t=0}^{T-1} \mathbb{E}\left[\left(1 - \frac{\tilde{\eta}\mu_F}{2}\right)\frac{\alpha_t \Delta_t}{\tilde{\eta} A_T} - \frac{\alpha_t \Delta_{t+1}}{\tilde{\eta} A_T}\right] + \frac{\tilde{\eta}^2}{\beta^2}C_3 + \tilde{\eta}C_2 + C_1$$

$$\leq \sum_{t=0}^{T-1} \mathbb{E}\left[\frac{\alpha_{t-1}\Delta_t - \alpha_t \Delta_{t+1}}{\tilde{\eta} A_T}\right] + \frac{\tilde{\eta}^2}{\beta^2}C_3 + \tilde{\eta}C_2 + C_1 \qquad (32)$$

$$= \frac{1}{\tilde{\eta} A_T}\Delta_0 - \frac{\alpha_{T-1}}{\tilde{\eta} A_T}\mathbb{E}\left[\Delta_T\right] + \frac{\tilde{\eta}^2}{\beta^2}C_3 + \tilde{\eta}C_2 + C_1$$

$$\leq \mu_F e^{-\tilde{\eta}\mu_F T/2}\Delta_0 - \frac{\mu_F}{2}\mathbb{E}\left[\Delta_T\right] + \frac{\tilde{\eta}^2}{\beta^2}C_3 + \tilde{\eta}C_2 + C_1, \qquad (33)$$

where we have (32) because in order for telescoping, we choose $\left(1 - \frac{\tilde{\eta}\mu_F}{2}\right)\alpha_t = \alpha_{t-1}$, and thus $\alpha_t = \left(1 - \frac{\tilde{\eta}\mu_F}{2}\right)^{-(t+1)}$ by recursive update. Regarding to (33), we have

$$A_T = \sum_{t=0}^{T-1}\left(1 - \frac{\tilde{\eta}\mu_F}{2}\right)^{-(t+1)}$$

$$= \left(1 - \frac{\tilde{\eta}\mu_F}{2}\right)^{-T}\sum_{t=0}^{T-1}\left(1 - \frac{\tilde{\eta}\mu_F}{2}\right)^{t}$$

$$= a_{T-1}\frac{1 - \left(1 - \frac{\tilde{\eta}\mu_F}{2}\right)^T}{\tilde{\eta}\mu_F/2}$$

which implies

$$\frac{a_{T-1}}{\tilde{\eta}\mu_F} \leq A_T \leq \frac{2a_{T-1}}{\tilde{\eta}\mu_F},$$

where the first inequality is due to the fact that $\left(1 - \frac{\tilde{\eta}\mu_F}{2}\right)^T \leq \exp(-\tilde{\eta}\mu_F T/2) \leq \exp(-1) \leq 1/2$ by setting $\tilde{\eta}T \geq \frac{2}{\mu_F}$ and the second inequality is due to $1 - \left(1 - \frac{\tilde{\eta}\mu_F}{2}\right)^T \leq 1$; thus we have $\frac{\alpha_{T-1}}{\tilde{\eta} A_T} \geq \frac{\mu_F}{2}$ and $\frac{1}{\tilde{\eta} A_T} \leq \mu_F\left(1 - \frac{\tilde{\eta}\mu_F}{2}\right)^T \leq \mu_F e^{-\tilde{\eta}\mu_F T/2}$.

Due to the convexity of $F(\cdot)$, (33) implies

$$\mathbb{E}\left[F\left(\sum_{t=0}^{T-1}\frac{\alpha_t}{A_T}w_t\right)\right] - F(w^*) + \frac{\mu_F}{2}\mathbb{E}\left[\Delta_T\right] \leq \mu_F\Delta_0 e^{-\tilde{\eta}\mu_F T/2} + \frac{\tilde{\eta}^2}{\beta^2}C_3 + \tilde{\eta}C_2 + C_1 \quad (34)$$

which implies

$$\mathbb{E}\left[F(\bar{w}_T) - F(w^*)\right] \leq \mu_F\Delta_0 e^{-\tilde{\eta}\mu_F T/2} + \frac{\tilde{\eta}^2}{\beta^2}C_3 + \tilde{\eta}C_2 + C_1. \qquad (35)$$

Next, using the techniques in [3–5], we consider following cases:

- If $\hat{\eta}_1 \geq \max\left\{\frac{2\ln\left(\mu_F^2\Delta_0 T/2C_2\right)}{\mu_F T}, \frac{2}{\mu_F T}\right\} =: \eta'$, then we choose $\tilde{\eta} = \eta'$; thus, having

$$\mathbb{E}\left[F(\bar{w}_T) - F(w^*)\right] \leq \mu_F\Delta_0 e^{-\ln\left(\mu_F^2\Delta_0 T/2C_2\right)} + \eta'C_2 + \frac{\eta'^2}{\beta^2}C_3 + C_1$$

$$\leq \tilde{\mathcal{O}}\left(\frac{C_2}{T\mu_F}\right) + \tilde{\mathcal{O}}\left(\frac{C_3}{T^2\beta^2\mu_F^2}\right) + C_1.$$

- If $\frac{2}{\mu_F T} \leq \hat{\eta}_1 \leq \frac{2\ln\left(\mu_F^2\Delta_0 T/2C_2\right)}{\mu_F T}$, then we choose $\tilde{\eta} = \hat{\eta}_1$; thus, having

$$\mathbb{E}\left[F(\bar{w}_T) - F(w^*)\right] \leq \mu_F\Delta_0 e^{-\hat{\eta}_1\mu_F T/2} + \tilde{\mathcal{O}}\left(\frac{C_2}{T\mu_F}\right) + \tilde{\mathcal{O}}\left(\frac{C_3}{T^2\beta^2\mu_F^2}\right) + C_1.$$

Combining two cases, we obtain

$$\mathbb{E}\left[F(\bar{w}_T) - F(w^*)\right] \le \mathcal{O}\big(\mathbb{E}\left[F(\bar{w}_T) - F(w^*)\right]\big) :=$$

$$\mathcal{O}\Big(\Delta_0\mu_F e^{-\hat{\eta}_1\mu_F T/2}\Big) + \tilde{\mathcal{O}}\bigg(\frac{(N/S-1)\sigma_{F,1}^2}{\mu_F TN}\bigg) + \tilde{\mathcal{O}}\bigg(\frac{(R\sigma_{F,1}^2 + \delta^2\lambda^2)\kappa_F}{R(T\beta\mu_F)^2}\bigg) + \mathcal{O}\bigg(\frac{\lambda^2\delta^2}{\mu_F}\bigg),$$

which finishes the proof of part (a). We next prove part (b) as follows

$$\mathbb{E}\left[\left\|\tilde{\theta}_i^T(w_T) - w^*\right\|^2\right]$$

$$\le 3\,\mathbb{E}\left[\left\|\tilde{\theta}_i^T(w_T) - \hat{\theta}_i^T(w_T)\right\|^2 + \left\|\hat{\theta}_i^T(w_T) - w_T\right\|^2 + \left\|w_T - w^*\right\|^2\right]$$

$$\le 3\Big(\delta^2 + \frac{1}{\lambda^2}\mathbb{E}\left[\left\|\nabla F_i(w_T)\right\|^2\right] + \mathbb{E}\left[\left\|w_T - w^*\right\|^2\right]\Big)$$

$$\le 3\Big(\delta^2 + \frac{2}{\lambda^2}\mathbb{E}\left[\left\|\nabla F_i(w_T) - \nabla F_i(w^*)\right\|^2 + \left\|\nabla F_i(w^*)\right\|^2\right] + \mathbb{E}\left[\left\|w_T - w^*\right\|^2\right]\Big)$$

$$\le 3\Big(\delta^2 + 3\mathbb{E}\left[\left\|w_T - w^*\right\|^2\right] + \frac{2}{\lambda^2}\left\|\nabla F_i(w^*)\right\|^2\Big),$$

where the last inequality is due to smoothness of $F_i$ with $L_F = \lambda$ according to Proposition 1. Take the average over $N$ clients, we have

$$\frac{1}{N}\sum_{i=1}^{N}\mathbb{E}\left[\left\|\tilde{\theta}_i^T(w_T) - w^*\right\|^2\right] \le 9\,\mathbb{E}\left[\left\|w_T - w^*\right\|^2\right] + \frac{6\sigma_{F,1}^2}{\lambda^2} + 3\delta^2$$

$$\le \frac{1}{\mu_F}\mathcal{O}\big(\mathbb{E}\left[F(\bar{w}_T) - F(w^*)\right]\big) + \mathcal{O}\bigg(\frac{\sigma_{F,1}^2}{\lambda^2} + \delta^2\bigg),$$

where the last inequality is by using (35) and (36), we can easily obtain

$$\mathbb{E}\left[\left\|w_T - w^*\right\|^2\right] \le \frac{2}{\mu_F}\bigg(\mu_F\Delta_0 e^{-\tilde{\eta}\mu_F T/2} + \frac{\tilde{\eta}^2}{\beta^2}C_3 + \tilde{\eta}C_2 + C_1\bigg)$$

$$= \frac{1}{\mu_F}\mathcal{O}\big(\mathbb{E}\left[F(\bar{w}_T) - F(w^*)\right]\big).$$

$\square$

## A.5   Theorem 2

*Proof.* We first prove part (a). Due to the $L_F$-smoothness of $F(\cdot)$, we have

$$\mathbb{E}\left[F(w_{t+1}) - F(w_t)\right]$$

$$\le \mathbb{E}\left[\langle\nabla F(w_t), w_{t+1} - w_t\rangle\right] + \frac{L_F}{2}\mathbb{E}\left[\left\|w_{t+1} - w_t\right\|^2\right]$$

$$= -\tilde{\eta}\mathbb{E}\left[\langle\nabla F(w_t), g_t\rangle\right] + \frac{\tilde{\eta}^2 L_F}{2}\mathbb{E}\left[\left\|g_t\right\|^2\right]$$

$$= -\tilde{\eta}\mathbb{E}\left[\left\|\nabla F(w_t)\right\|^2\right] - \tilde{\eta}\mathbb{E}\left[\langle\nabla F(w_t), g_t - \nabla F(w_t)\rangle\right] + \frac{\tilde{\eta}^2 L_F}{2}\mathbb{E}\left[\left\|g_t\right\|^2\right]$$

$$\le -\tilde{\eta}\mathbb{E}\left[\left\|\nabla F(w_t)\right\|^2\right] + \frac{\tilde{\eta}}{2}\mathbb{E}\left[\left\|\nabla F(w_t)\right\|^2\right] + \frac{\tilde{\eta}}{2}\mathbb{E}\left\|\frac{1}{NR}\sum_{i,r}^{N,R} g_{i,r}^t - \nabla F_i(w_t)\right\|^2 + \frac{\tilde{\eta}^2 L_F}{2}\mathbb{E}\left[\left\|g_t\right\|^2\right]$$

$$\tag{36}$$

$$\le -\frac{\tilde{\eta}}{2}\mathbb{E}\left[\left\|\nabla F(w_t)\right\|^2\right] + \frac{3L_F\tilde{\eta}^2}{2}\mathbb{E}\left\|\frac{1}{S}\sum_{i\in\mathcal{S}^t}\nabla F_i(w_t) - \nabla F(w_t)\right\|^2$$

$$+ \frac{\tilde{\eta}(1 + 3L_F\tilde{\eta})}{2}\frac{1}{NR}\sum_{i,r}^{N,R}\mathbb{E}\left[\left\|g_{i,r}^t - \nabla F_i(w_t)\right\|^2\right] + \frac{3\tilde{\eta}^2 L_F}{2}\mathbb{E}\left[\left\|\nabla F(w_t)\right\|^2\right] \tag{37}$$

$$\le -\frac{\tilde{\eta}(1 - 3L_F\tilde{\eta})}{2}\mathbb{E}\left[\left\|\nabla F(w_t)\right\|^2\right] + \frac{3L_F\tilde{\eta}^2}{2}\frac{N/S-1}{N-1}\sum_{i=1}^{N}\frac{1}{N}\mathbb{E}\left[\left\|\nabla F_i(w_t) - \nabla F(w_t)\right\|^2\right]$$

$$+ \frac{\tilde{\eta}(1 + 3L_F\tilde{\eta})}{2}\left[2\lambda^2\delta^2 + \frac{16\tilde{\eta}^2 L_F^2}{\beta^2}\left(\frac{2\lambda^2\delta^2}{R} + 3\sum_{i=1}^{N}\frac{1}{N}\mathbb{E}\left[\|\nabla F_i(w_t) - \nabla F(w_t)\|^2\right] + 3\mathbb{E}\left[\|\nabla F(w_t)\|^2\right]\right)\right]$$

<div align="right">(38)</div>

$$\leq -\frac{\tilde{\eta}(1 - 3L_F\tilde{\eta})}{2}\mathbb{E}\left[\|\nabla F(w_t)\|^2\right] + \frac{3L_F\tilde{\eta}^2}{2}\frac{N/S - 1}{N - 1}\left(\sigma_{F,2}^2 + \frac{8L^2}{\lambda^2 - 8L^2}\mathbb{E}\left[\|\nabla F(w_t)\|^2\right]\right)$$

$$+ \frac{\tilde{\eta}(1 + 3L_F\tilde{\eta})}{2}\left[2\lambda^2\delta^2 + \frac{16\tilde{\eta}^2 L_F^2}{\beta^2}\left(\frac{2\lambda^2\delta^2}{R} + 3\sigma_{F,2}^2 + \frac{3\lambda^2}{\lambda^2 - 8L^2}\mathbb{E}\left[\|\nabla F(w_t)\|^2\right]\right)\right] \quad (39)$$

$$= -\frac{\tilde{\eta}(1 - 3L_F\tilde{\eta})}{2}\mathbb{E}\left[\|\nabla F(w_t)\|^2\right] + \tilde{\eta}^2 L_F\left(\frac{12L^2}{\lambda^2 - 8L^2}\frac{N/S - 1}{N - 1} + \frac{24\tilde{\eta}(1 + 3L_F\tilde{\eta})\lambda^2 L_F}{\beta^2(\lambda^2 - 8L^2)}\right)\mathbb{E}\left[\|\nabla F(w_t)\|^2\right]$$

$$+ \frac{\tilde{\eta}^3}{\beta^2}(1 + 3L_F\tilde{\eta})\frac{8(3R\sigma_{F,2}^2 + 2\delta^2\lambda^2)}{R} + \tilde{\eta}^2\sigma_{F,2}^2\left(\frac{3L_F}{2}\frac{N/S - 1}{N - 1}\right) + \tilde{\eta}(1 + 3L_F\tilde{\eta})\lambda^2\delta^2$$

<div align="right">(40)</div>

$$\leq -\tilde{\eta}\underbrace{\left[1 - \tilde{\eta}L_F\left(\frac{3}{2} + \frac{12L^2}{\lambda^2 - 8L^2}\frac{N/S - 1}{N - 1} + \frac{36\lambda^2}{\lambda^2 - 8L^2}\right)\right]}_{\geq 1/2 \text{ when } \tilde{\eta} \text{ satisfied } (44)}\mathbb{E}\left[\|\nabla F(w_t)\|^2\right]$$

$$+ \frac{\tilde{\eta}^3}{\beta^2}(1 + 3L_F\tilde{\eta})\frac{8(3R\sigma_{F,2}^2 + 2\delta^2\lambda^2)}{R} + \tilde{\eta}^2\frac{3L_F\sigma_{F,2}^2}{2}\frac{N/S - 1}{N - 1} + \tilde{\eta}(1 + 3L_F\tilde{\eta})\lambda^2\delta^2 \quad (41)$$

$$\leq -\frac{\tilde{\eta}}{2}\|\nabla F(w_t)\|^2 + + \frac{\tilde{\eta}^3}{\beta^2}\underbrace{\frac{16(3R\sigma_{F,2}^2 + 2\delta^2\lambda^2)}{R}}_{=:C_4} + \tilde{\eta}^2\underbrace{\frac{3L_F\sigma_{F,2}^2}{2}\frac{N/S - 1}{N - 1}}_{=:C_5} + \tilde{\eta}\underbrace{2\lambda^2\delta^2}_{=:C_6} \quad (42)$$

where (37) is due to Cauchy-Swartz and AM-GM inequalities, (38) is by decomposing $\|g_t\|^2$ into three terms according to (18), and (39) is by using Lemmas 4 and 5, and the fact that $\mathbb{E}\left[\|X\|^2\right] = \mathbb{E}\left[\|X - \mathbb{E}[X]\|^2\right] + \mathbb{E}[\|X\|]^2$ for any vector of random variable $X$. We have (40) by Lemma 2, and (41) by re-arranging the terms, and (42) by having $1 + 3L_F\tilde{\eta} \leq 1 + \frac{3\beta}{2} \leq 3\beta$ when $\tilde{\eta} \leq \frac{\beta}{2L_F}$ according to Lemma 5 and $\beta \geq 1$. Finally, we have (43) by using the condition $\lambda^2 - 8L^2 \geq 1$ and the fact that $\frac{N/S - 1}{N - 1} \leq 1$ for the following

$$L_F\left(\frac{3}{2} + \frac{12L^2}{\lambda^2 - 8L^2}\frac{N/S - 1}{N - 1} + \frac{36\lambda^2}{\lambda^2 - 8L^2}\right) \leq \frac{L_F}{2}\left(3 + 24L^2 + 72\lambda^2\right) \leq \frac{L_F}{2}\left(75\lambda^2\right)$$

to get

$$1 - \tilde{\eta}L_F\left(\frac{3}{2} + \frac{12L^2}{\lambda^2 - 8L^2}\frac{N/S - 1}{N - 1} + \frac{36\lambda^2}{\lambda^2 - 8L^2}\right) \geq 1 - \frac{75\tilde{\eta}L_F\lambda^2}{2} \geq \frac{1}{2}$$

with the condition

$$\tilde{\eta} \leq \frac{1}{75L_F\lambda^2} =: \hat{\eta}_2, \quad (43)$$

which also implies $1 + 3L_F\tilde{\eta} \leq 1 + \frac{1}{25\lambda^2} \leq 2$.

We note that $\hat{\eta}_2 \leq \frac{\beta}{2L_F}$ with $\beta \geq 1$. By re-arranging the terms of (43) and telescoping, we have

$$\frac{1}{2T}\sum_{t=0}^{T-1}\mathbb{E}\left[\|\nabla F(w_t)\|^2\right] \leq \frac{\mathbb{E}\left[F(w^0) - F(w_T)\right]}{\tilde{\eta}T} + \frac{\tilde{\eta}^2}{\beta^2}C_4 + \tilde{\eta}C_5 + C_6. \quad (44)$$

Defining $\Delta_F := F(w^0) - F^*$, and following the techniques used by [3–5], we consider two cases:

- If $\hat{\eta}_2^3 \geq \frac{\beta^2\Delta_F}{TC_4}$ or $\hat{\eta}_2^2 \geq \frac{\Delta_F}{TC_5}$, then we choose $\tilde{\eta} = \min\left\{\left(\frac{\beta^2\Delta_F}{TC_4}\right)^{\frac{1}{3}}, \left(\frac{\Delta_F}{TC_5}\right)^{\frac{1}{2}}\right\}$; thus, having

$$\frac{1}{2T}\sum_{t=1}^{T-1}\mathbb{E}\left[\|\nabla F(w_t)\|^2\right] \leq \frac{(\Delta_F)^{2/3}C_4^{1/3}}{(\beta^2 T)^{2/3}} + \frac{(\Delta_F C_5)^{1/2}}{\sqrt{T}} + C_6.$$

<div align="center">9</div>

- If $\hat{\eta}_2^3 \leq \frac{\beta^2 \Delta_F}{T C_4}$ and $\hat{\eta}_2^2 \leq \frac{\Delta_F}{T C_5}$, then we choose $\tilde{\eta} = \hat{\eta}_2$. We have

$$\frac{1}{2T} \sum_{t=0}^{T-1} \mathbb{E}\left[\|\nabla F(w_t)\|^2\right] \leq \frac{\Delta_F}{\hat{\eta}_2 T} + \frac{(\Delta_F)^{2/3}(C_4)^{1/3}}{(\beta^2 T)^{2/3}} + \frac{(\Delta_F C_5)^{1/2}}{\sqrt{T}} + C_6.$$

Combining two cases, and with $t^*$ uniformly sampled from $\{0, \ldots, T-1, \}$ we have

$$\frac{1}{T} \sum_{t=0}^{T-1} \mathbb{E}\left[\|\nabla F(w_t)\|^2\right] = \mathbb{E}\left[\|\nabla F(w_{t^*})\|^2\right] \leq \mathcal{O}\Big(\mathbb{E}\left[\|\nabla F(w_{t^*})\|^2\right]\Big) :=$$

$$\mathcal{O}\left(\frac{\Delta_F}{\hat{\eta}_2 T} + \frac{(\Delta_F)^{\frac{2}{3}}\left(R\sigma_{F,2}^2 + \lambda^2\delta^2\right)^{\frac{1}{3}}}{\beta^{\frac{4}{3}} R^{\frac{1}{3}} T^{\frac{2}{3}}} + \frac{\left(\Delta_F L_F \sigma_{F,2}^2 (N/S - 1)\right)^{\frac{1}{2}}}{\sqrt{TN}} + \lambda^2\delta^2\right)$$

which proves the first part of Theorem 2.

We next prove part (b) as follows

$$\frac{1}{N} \sum_{i=1}^{N} \mathbb{E}\left[\|\tilde{\theta}_i^t(w_t) - w_t\|^2\right] \leq \frac{1}{N} \sum_{i=1}^{N} 2\mathbb{E}\left[\|\tilde{\theta}_i^t(w_t) - \hat{\theta}_i^t\|^2 + \|\hat{\theta}_i^t(w_t) - w_t\|^2\right]$$

$$\leq 2\delta^2 + \frac{2}{N} \sum_{i=1}^{N} \frac{\mathbb{E}\left[\|\nabla F_i(w_t)\|^2\right]}{\lambda^2}$$

$$\leq 2\delta^2 + \frac{2}{\lambda^2 - 8L^2} \mathbb{E}\left[\|\nabla F(w_t)\|^2\right] + \frac{2\sigma_{F,2}^2}{\lambda^2}, \qquad (45)$$

where the first inequality is due to Proposition (3) and the third inequality is by using the fact that $\mathbb{E}\left[\|X\|^2\right] = \mathbb{E}\left[\|X - \mathbb{E}[X]\|^2\right] + \mathbb{E}[\|X\|]^2$ for any vector of random variable $X$, we have

$$\frac{1}{N} \sum_{i=1}^{N} \mathbb{E}\left[\|\nabla F_i(w_t)\|^2\right] = \sum_{i=1}^{N} \frac{1}{N} \Big(\mathbb{E}\left[\|\nabla F_i(w_t) - \nabla F(w_t)\|^2\right] + \mathbb{E}\left[\|\nabla F(w_t)\|^2\right]\Big)$$

$$\leq \sigma_{F,2}^2 + \frac{\lambda^2}{\lambda^2 - 8L^2} \mathbb{E}\left[\|\nabla F(w_t)\|^2\right].$$

Summing (**??**) from $t = 0$ to $T$, we get

$$\frac{1}{TN} \sum_{i=0}^{T-1} \sum_{i=1}^{N} \mathbb{E}\left[\|\tilde{\theta}_i^t - w_t\|^2\right] \leq \frac{2}{\lambda^2 - 8L^2} \frac{1}{T} \sum_{t=0}^{T-1} \mathbb{E}\left[\|\nabla F(w_t)\|^2\right] + 2\delta^2 + \frac{2\sigma_{F,2}^2}{\lambda^2},$$

and with $t^*$ uniformly sampled from $\{0, \ldots, T-1\}$, we finish the proof. $\qquad \square$

# B  Additional Experimental Settings And Results

## B.1  Additional Experimental Environment Settings

We implemented `pFedMe`, FedAvg, and Per-FedAvg using PyTorch [6] and run the experiments on multiple computers using the Intel Core i7-9700K CPU and 32GB of RAM. Each experiment is run at least 10 times for statistical reports.

## B.2  Effect of hyperparameters

To understand how different hyperparameters such as $R$, $|\mathcal{D}|$, and $\lambda$ affect the convergence of `pFedMe` in both $\mu$-strongly convex and nonconvex settings, we conduct various experiments on MNIST dataset with $\eta = 0.005$ and $S = 5$.

**Effects of local computation rounds** $R$: When the communication is relatively costly, the server tends to allow users to have more local computations, which can lead to less global model updates and thus faster convergence. Therefore, we monitor the behavior of `pFedMe` using a number of

Figure 1: Effect of $R$ on the convergence of pFedMe in $\mu$-strongly convex and nonconvex settings on MNIST ($|\mathcal{D}| = 20$, $\lambda = 15$, $K = 5$, $\beta = 1$).

Figure 2: Effect of $|\mathcal{D}|$ on the convergence of pFedMe in $\mu$-strongly convex and nonconvex settings on MNIST ($\lambda = 15$, $R = 20$, $K = 5$, $\beta = 1$).

Figure 3: Effect of $\lambda$ on the convergence of pFedMe in $\mu$-strongly convex and nonconvex settings on MNIST ($|\mathcal{D}| = 20$, $R = 20$, $K = 5$, $\beta = 1$).

values of $R$, which results in Fig. 1. The results show that larger values of $R$ have a benefit on the convergence of both the personalized and the global models. There is, nevertheless, a trade-off between the computations and communications: while larger $R$ requires more computations at local users, smaller $R$ needs more global communication rounds to converge. To balance this trade-off, we fix $R = 20$ and evaluate the effect of other hyperparameters accordingly.

**Effects of Mini-Batch size** $|\mathcal{D}|$: As mentioned in the Lemma 1, $|\mathcal{D}|$ is one of the parameters which can be controlled to adjust the value of $\delta$. In Fig. 3, when the size of the mini-batch is increased, pFedMe has the higher convergence rate. However, very large $|\mathcal{D}|$ will not only slow the convergence of pFedMe but also requires higher computations at the local users. During the experiments, the value of $|\mathcal{D}|$ is configured as a constant value equal to 20.

**Effects of regularization** $\lambda$: Fig. 4 shows the convergence rate of pFedMe with different values of $\lambda$. In all settings, larger $\lambda$ allows for faster convergence; however, we also observe that the significantly large $\lambda$ will hurt the performance of pFedMe by making pFedMe diverge. Therefore, $\lambda$ should be tuned carefully depending on the dataset. We fix $\lambda = 15$ for all scenarios with MNIST.