[Reviews · NeurIPS 2020]

Review 1

Summary and Contributions: This paper presents a nice formulation of a federated learning with personalization objective, where a the average of the Moreau envelopes of the local objectives is minimized, and the argmin of the Moreau envelope / prox is used as the personalized weights for each local objective. This formulation has advantages over previously considered ones, and a straightforward FedAvg analogue is analyzed and shown to converge in a certain sense (see below). ------- update after author response ---------- Thank you for addressing many of my comments. I understand that you can't separate the (lambda / 2)\| theta_i - w \|^2 terms from the rest in the objective, just like with L2 regularization. The point I was making was that you should separate between the algorithm and the goal. For instance, ridge regression is an algorithm which you might use for least squares, but at the end of the day, you would report the mean squared error and you would *not* report the mean squared error plus the regularization term. The regularization was just part of the algorithm, not the goal. It seems to me that the (lambda / 2)\| theta_i - w \|^2 is analogous to a regularization term here, and is not really the thing you want to make small itself. It is true that you can't choose lambda=0 for your method, but you can choose lambda=exp(-100000) and your theory does say that smaller lambda is better. However, I suspect that there is some perspective on the problem where choosing lambda>>0 can be shown to be advantageous (just as choosing lambda>>0 can be for ridge regression). I think it would be valuable to identify this perspective.

Strengths: I like the formulation of the objective, which seems like a good conceptual balance between fully personalizing each local model and still leveraging information from other machines at the same time. The algorithm is simple and natural, and the experimental results show that this method appears to have significant practical benefits.

Weaknesses: I am not completely convinced by the theoretical results. Specifically, I am not sure that Theorems 1 and 2 prove the right notion of convergence. Conceptually, I think you want to show that sum_i f_i(theta_i) is small and/or sum_i f_i(w) is small. The theorem statements, I suppose, upper bound sum_i f_i(theta_i), but \| w - theta_i \| is involved, and I don't see why that quantity should be relevant. I don't really see any reason to care about \| w - theta_i \|; so what if you need to move far away from w to optimally personalize for one particular objective? In short, I think that the proposed objective pFedMe makes sense as a *training/surrogate objective*, but does not make as much sense as a criterion for evaluating a model, if that makes sense. I also think that an important consideration, which is missing here, is figuring out to what extent you need to leverage information from other clients in order to achieve optimal performance. On the one hand, choosing lambda=0 means just setting theta_i to the ERM on the local objective, which might do alright, but does not leverage the data / similarity with other clients. On the other hand, setting lambda = \infty allows for no personalization at all. Presumably, something in between is ideal, or else there is no reason for your formulation/method! According to Theorem 1, to the extent that I want to minimize F(w) - F*, I should just set lambda = 0 (since the rhs decreases with lambda). However, that is pretty boring since it means you should just do ERM locally on each client. I think the theory could be greatly improved by: (a) directly analyzing sum_i f_i(theta_i) and perhaps also sum_i f_i(w), i.e. the actual objective of interest (b) showing that increasing lambda can decrease f_i(theta_i) (perhaps this would require additional assumptions about the objectives) In the experiments, I am not completely clear with what is being plotted for the personalized/global models. Is the personalized model plotting average across the local objectives of the loss on the personalized parameter theta_i, and the global model is the average across the local objectives of the loss of the global parameter w? Perhaps I missed something, but I don't think this is precisely explained. Also, for the experiments, assuming I was correct in my interpretation above, I think it would also be good to show the performance of the personalized models for Per-FedAvg (i.e. the local weights after taking an SGD step from the global params). I realize this might not make a huge difference, but it seems like the right thing to compare with pFedMe (PM).

Correctness: To my knowledge, the results are correct.

Clarity: The paper was clearly presented, see above.

Relation to Prior Work: This paper discusses its relationship to prior work. Since the formulation of the objective is changed, it is somewhat difficult to compare convergence rates between this and prior methods (e.g. PerFedAvg), however a more explicit comparison might be helpful here. In particular, I am not sure how to understand the sentence lines 201-203 which says that prior methods can't get quadratic speedup--presumably those methods analyzed a different objective (maybe like eq (1))? More explanation here would help.

Reproducibility: Yes

Additional Feedback:


Review 2

Summary and Contributions: This paper tackles the task of federated learning on non-iid data distributions over the agents. The approach focuses on local models, with a additional regularization term enforcing proximity to a consensus model. The chosen regularization is a quadratic term, leading to a convenient reformulation as a distributed proximal problem. Detailed theoretical bounds are provided as well as several empirical evidences.

Strengths: The paper is very well written: the problem is clearly stated, the proposed solution is well motivated and analyzed. Provided results seem encouraging regarding the relevance of the algorithm.

Weaknesses: This problem has already been investigated in decentralized frameworks. Although different from federated learning, the proximity of the methods in both domains should be discussed (see, e.g., Decentralized collaborative learning of personalized models over networks, P Vanhaesebrouck, A Bellet, M Tommasi - 2017)

Correctness: The claims are correct as well as the overall analysis. At line 167, the author chooses a parameter to ensure strong convexity of h. However, the assumption on f_i is done in expectation: in some cases where the subset is poorly drawn, would not \lambda be very large to guarantee this property? What would be the dependence of \lambda in the expectation/variance term to ensure such property with high probability?

Clarity: The paper is very well written. Some bounds contain a lot of details, but simpler corollary and extensive discussions are provided.

Relation to Prior Work: Related work is well discussed, both in the dedicated section and when discussing results. As mentioned earlier, maybe some additional discussions comparing federated and decentralized results would be enlightening.

Reproducibility: Yes

Additional Feedback: I took some time to browse the supplementary material but I did not have time to carefully read it entirely, it is possible I missed something answering my concerns. I would be happy to modify my review if such event occurred.


Review 3

Summary and Contributions: This paper introduces a novel personalised federated learning algorithm, pFedMe, which relies on Moreau envelopes. The authors prove convergence bounds in the strongly convex and smooth non-convex settings. Experiments on real and synthetic data in the convex (logistic regression) and non-convex (shallow neural network) settings demonstrate improved performance with respect to state-of-the-art.

Strengths: - This paper reformulates previous personalisation ideas stemming from meta-learning approaches (notably Per-FedAvg) in the well-studied proximal framework. The new algorithm is more generic (agnostic to the local optimiser) and at the same time simpler to understand and analyse. This is a significant and novel contribution. It is very relevant to the NeurIPS community, as reformulations help to generate new ideas. - The convergence analysis relies on standard assumptions (avoiding the idealistic gradient boundedness) and yields state-of-the-art results. - Experiments on real and synthetic data show better results than previous personalised algorithms and FedAvg, and are quite compelling.

Weaknesses: The convergence analysis seems to rely strongly on a very particular server aggregation with a lag term controlled by a beta term. In particular, beta needs to be much larger than 1 to reach the SOTA speedup (Corollaries 1 and 2). Intuitively, the update rule with beta larger than 1 could yield some instabilities, and indeed. one observes in Fig 5 from the appendix that a larger beta yields a more instable optimisation in the strongly convex case. The authors state that the learning rate must be tuned very carefully in this case. The drawback is that this brings yet another hyper parameter to tune, which is costly in the FL setting.

Correctness: - The theoretical claims seem correct. - Regarding the empirical methodology, the proposed algorithm used K*R local steps instead of R steps for the other algorithms. I would have appreciated a comparison of their algorithm with fedAvg and per-FedAvg run with K*R local steps, to better understand if the improved performance comes from the additional local computations or from the inherent better formulation. - It would also have been better to use a more realistic FL dataset than MNIST, for instance the LEAF benchmark. - If I understand things clearly, all experimental performances are means of local test performances. How are these averages computed? Is there a weighting with respect to the number of samples? How do the local test performance distributions look like with the different algorithms? - It is unclear whether the hyper parameters were optimised on the local test sets or on local validation sets.

Clarity: The paper is clear and well written. I have spotted very few typos (see additional comments).

Relation to Prior Work: The authors discuss well previous work overall.

Reproducibility: Yes

Additional Feedback: - The global model of pFedMe is lagging in the synthetic experiments (Fig 2), whereas it’s comparable to FedAvg in MNIST (Fig 1). Any idea why it is the case? Is it due to the varying number of clients? Or to a lower sampling rate (S/N = 10% vs 50% ? - Line 175, typo: let theta « is » -> « be » - Line 179, « let assumption 1(a) hold » (no s

[Author Response · NeurIPS 2020]

We thank the reviewers for valuable feedbacks. In below response, the reference order is the same as submitted work.

**Reviewer # 1:**

*1) The convergence theory should be w.r.t $\sum_i f_i(\theta_i)$ and/or $\sum_i f_i(w)$, why care about $\|w - \theta_i\|$?* $\rightarrow$ pFedMe can be written compactly as $\min_w \min_{\theta_i}(1/N)\sum_{i=1}^{N}\{f_i(\theta_i) + (\lambda/2)\|\theta_i - w\|^2\}$, which shows that there is dependency between solution components $\{\theta_i\}_i$ and $w$. Thus, we cannot separately evaluate each objective term. Similar to, e.g., $l_2$-regularization where $f_i(w) = l_i(w) + (\lambda/2)\|w\|^2$ for some loss function $l_i(w)$ and the overfitting regularization term $\|w\|^2$ plays a critical role in convergence of $f_i$, we believe that it makes sense to consider the personalized regularization $\|\theta_i - w\|^2$ in pFedMe's convergence analysis. If we want to move far away from $w$, setting $\lambda$ small is one option.

*2) To what extent of $\lambda$ to leverage client aggregation? From Theorem 1, $\lambda = 0$ to minimize $F(w) - F^*$?* $\rightarrow$ The trade-off between personalization and exploiting data aggregation by varying $\lambda$ is skilfully phrased by the reviewer. Note that $\lambda \in (0, \infty)$ avoids extreme cases of $\lambda = 0$ (no FL) and $\lambda = \infty$ (no personalization). The reviewer is correct in that setting $\lambda = 0$ reduces to simple ERM on each local client, but it is not true that in order to minimize $F(w) - F^*$, we should just set $\lambda = 0$, because Theorem 1 is based on using the Moreau envelope $F_i$, which implicitly requires $\lambda > 0$ by its definition [39]. In other words, if $\lambda = 0$, there would be no local update in line 8, and hence no Theorem 1 (and 2). On the other hand, similar to the standard way to find a "sweet spot" for hyper-parameters in $l_1$- or $l_2$-regularization, we fine-tune $\lambda$ based on each dataset for each task in our experiments.

*3) Are personalized model and global model plottings w.r.t $\theta_i$ and $w$, respectively?* $\rightarrow$ Yes.

*4) Use personalized models for Per-FedAvg's performance?* $\rightarrow$ We did use $\theta_i(w) = w - \alpha\nabla f_i(w)$ for Per-FedAvg.

*5) How to compare convergence rates with prior methods when they have different objectives?* $\rightarrow$ The standard convergence analysis is performed according to the **loss functions' properties such as strong/non-convexity and/or smoothness** [49], regardless of different learning taks/objectives. Therefore, the quadratic speedup of pFedMe is compared with other linear-speedup methods having the same strong convexity and smoothness properties [3, 6, 19].

**Reviewer #2:**

*1) Discuss with decentralized frameworks of Vanhaesebrouck, Bellet, Tommasi (2017)* $\rightarrow$ We will update it.

*2) How to choose $\lambda$ to ensure strong convexity of $h$, considering the assumption on $f_i$ is in expectation?* $\rightarrow$ Assumption 1 can apply a stronger $L$-smoothness assumption on $\tilde{f}_i(\theta_i; \xi_i), \forall \xi_i$, instead of $f_i$. In this case, we can choose arbitrary $\lambda > 0$ for strongly convex and $\lambda > L$ for nonconvex loss functions, respectively.

**Reviewer #4:**

*1) $\beta$ larger than 1 could yield some instabilities (e.g., Fig 5) and inducing one more costly hyperparameter to tune?* $\rightarrow$ In Theorem 1, we need $\eta \le \frac{\hat{\eta}_1}{\beta R}$, which means larger $\beta$ requires **smaller** $\eta$ for the stability. In practice, we can fix $\beta$ to an arbitrary constant, say 3, and perform the simple fine-tuning task for $\eta$.

| Algorithm | MNIST | Synthetic |
|---|---|---|
| FedAvg | $94.11 \pm 0.05$ | $77.69 \pm 0.2$ |
| Per-FedAvg | $94.22 \pm 0.02$ | $79.79 \pm 0.09$ |
| pFedMe-GM | $94.18 \pm 0.06$ | $78.65 \pm 0.25$ |
| pFedMe-PM | $\mathbf{95.62} \pm 0.04$ | $\mathbf{83.20} \pm 0.06$ |

Figure 1: Accuracy comparison with fine-tuned hyperparameters. $R = 20$, $K = 5$ for pFedMe, $R = 100$ for others.

*2) Run $K * R$ local steps for Per-FedAvg and FedAvg.* $\rightarrow$ It would be unfair to compare pFedMe with these using $K * R$ local steps. This is because for every $K$ steps in a local round $r$, pFedMe only uses a single mini-batch, and thus only **R** mini-batches during $R$ local rounds. On the other hand, Per-FedAvg uses 2 different mini-batches for 1 local update, and thus after $K * R$ local rounds, it will perform $\mathbf{2 * K * R}$ mini-batch updates. Similarly, FedAvg uses one mini-batch for 1 local update, so in total it will have $\mathbf{K * R}$ mini-batch updates. However, we still report an additional comparison when FedAvg and Per-FedAvg were trained over $K * R$ local steps in Fig. 1 here (in a strongly convex setting), which shows the personalized model (PM) of pFedMe still outperforms others.

*3) Use a more realistic FL dataset than MNIST (LEAF benchmark)?* $\rightarrow$ For synthetic data, we used a similar method to LEAF to generate the data. For real data, in LEAF, FEMNIST is distributed to a large number of clients, each with a small local dataset (of size around 226), which is not suitable for pFedMe because it needs a larger local dataset on each client (due to fresh mini-batch sampling in every round in line 7). Thus we use MNIST with a smaller number of clients, so each client can have up to 3,834 data samples. We note that the way we generate non-i.i.d and heterogeneous data using MNIST is similar to that using FEMNIST in LEAF.

*4) How do (local) test performances computed and distributed? Any weighting?* $\rightarrow$ W.r.t test accuracy of all algorithms, we sum all correctly classified samples over clients and divide it by the total number of samples of all clients without weighting. Their histograms are quite similar and we reported their means and standard deviation in Table 1.

*5) The hyper parameters were optimised on the local test sets or on local validation sets?* $\rightarrow$ On local test set. We have additionally fine-tuned hyper-parameters on a validation set, and still obtain the same values as those on the test set.

*6) Why is global model of pFedMe is lagging in Fig. 2? Due to varying number of clients or lower sampling rate?* $\rightarrow$ The **non-fine-tuned hyper-parameter** is the main reason for the lagging performance of the global model (GM) in Fig. 2. In cases where hyperparameters are fine-tuned, the GM is consistent and performs well compared to FedAvg (see Table 1). We have run more experiments and observed that the GM performance is still consistent with varying numbers of clients and sampling rates. The result is not presented here due to the lack of space.

*7)* Finally, we thank the reviewer for correcting several typos.

[Meta-Review · NeurIPS 2020]

There was a consensus among reviewers that this paper is sound and makes interesting contributions towards more personalized federated learning. After the author rebuttal, there was some discussion among reviewers regarding the "true" underlying objective that this approach is optimizing. It was decided that although very interesting, this point is beyond the scope of the paper. Therefore we accepted the paper, but I recommend that the authors add to the final version a discussion of potential future work along the lines of showing under which conditions the objective they propose can be provably useful (e.g., to improve generalization performance).